# Tissue-infiltrating macrophages mediate an exosome-based metabolic reprogramming upon DNA damage

Evi Goulielmaki[1,7], Anna Ioannidou[1,2,7], Maria Tsekrekou[1,2], Kalliopi Stratigi [1], Ioanna K. Poutakidou[1], Katerina Gkirtzimanaki[1], Michalis Aivaliotis [1], Konstantinos Evangelou[3], Pantelis Topalis[1], Janine Altmüller[4], Vassilis G. Gorgoulis [3,5,6], Georgia Chatzinikolaou[1] & George A. Garinis [1,2]*

DNA damage and metabolic disorders are intimately linked with premature disease onset but the underlying mechanisms remain poorly understood. Here, we show that persistent DNA damage accumulation in tissue-infiltrating macrophages carrying an ERCC1-XPF DNA repair defect ($Er1^{F/-}$) triggers Golgi dispersal, dilation of endoplasmic reticulum, autophagy and exosome biogenesis leading to the secretion of extracellular vesicles (EVs) in vivo and ex vivo. Macrophage-derived EVs accumulate in $Er1^{F/-}$ animal sera and are secreted in macrophage media after DNA damage. The $Er1^{F/-}$ EV cargo is taken up by recipient cells leading to an increase in insulin-independent glucose transporter levels, enhanced cellular glucose uptake, higher cellular oxygen consumption rate and greater tolerance to glucose challenge in mice. We find that high glucose in EV-targeted cells triggers pro-inflammatory stimuli via mTOR activation. This, in turn, establishes chronic inflammation and tissue pathology in mice with important ramifications for DNA repair-deficient, progeroid syndromes and aging.

[1] Institute of Molecular Biology and Biotechnology, Foundation for Research and Technology-Hellas, GR70013 Heraklion, Crete, Greece. [2] Department of Biology, University of Crete, Heraklion, Crete, Greece. [3] Department of Histology and Embryology, Athens Medical School, GR11527 Athens, Greece. [4] Cologne Center for Genomics (CCG), Institute for Genetics, University of Cologne, 50931 Cologne, Germany. [5] Biomedical Research Foundation of the Academy of Athens, 4 Soranou Ephessiou St. GR-11527, Athens, Greece. [6] Faculty of Biology, Medicine and Health, University of Manchester, Manchester Academic Health Science Centre, Wilmslow Road, Manchester M20 4QL, UK. [7] These authors contributed equally: Evi Goulielmaki, Anna Ioannidou. *email: garinis@imbb.forth.gr

To counteract DNA damage, mammalian cells have evolved partially overlapping DNA repair systems to remove DNA lesions and restore their DNA back to its native form[1,2]. For helix-distorting damage, cells employ the nucleotide excision repair (NER) pathway[3–5], a highly conserved mechanism that recognizes and removes helical distortions throughout the genome or selectively from the actively transcribed strand of genes[6,7]. In humans, defects in NER are causally linked to mutagenesis and cancer initiation as in the cancer-prone syndrome xeroderma pigmentosum (XP, complementation groups XP-A to XP-G)[8] or to developmental and neuronal abnormalities as seen in a heterogeneous group of progeroid syndromes, including the Cockayne syndrome (CS; affected genes: Csa, Csa), Trichothiodystrophy (TTD; affected genes: Xpb, Xpd) or the XPF-ERCC1 syndrome (XFE; affected genes: Ercc1, Xpf)[9–11]. The links between persistent DNA damage and the premature onset of age-related metabolic and endocrine perturbations in NER patients and accompanying mouse models[10,12–18] are well established. We and others have recently shown that chronic inflammation[19,20], genotoxic and oxidative stress[21–23] contribute significantly in NER progeria and age-related degenerative diseases, but the mechanisms remain unresolved[15,24,25]. XPF-ERCC1 is a highly conserved, heterodimeric, structure-specific endonuclease complex required for lesion excision in NER[26,27] that is thought to play an analogous role in the repair of DNA interstrand crosslinks (ICLs) that covalently link both DNA strands preventing transcription and replication[28].

Using mice with an engineered ERCC1-XPF defect in tissue-infiltrating macrophages, we provide evidence for a fundamental mechanism by which irreparable DNA damage triggers an exosome-based, metabolic reprogramming that leads to chronic inflammation and tissue pathology in NER progeroid syndromes and likely also during aging.

## Results

**Cytoplasmic stress responses in $Er1^{F/-}$ macrophages.** To dissect the functional links between irreparable DNA damage and innate immune responses in vivo, we intercrossed animals homozygous for the floxed Ercc1 allele ($Ercc1^{F/F}$)[29] with mice carrying the Lysozyme 2 (Lys2)-Cre transgene in an Ercc1 heterozygous background; Lys2 is a bacteriolytic enzyme that is primarily expressed in the monocyte-macrophage system[30]. Crossing the Lys2-Cre with Rosa YFP transgenic animals confirmed the specificity of Lys2-driven YFP expression to thioglycolate-elicited peritoneal macrophages (TEMs; Fig. 1a) but not to hepatocytes, the primary pancreatic cells (PPCs; Fig. 1b, c) or the pancreas and the white adipose tissue (WAT) that are infiltrated with MAC1-possitve macrophages expressing YFP (Fig. 1d). Western blotting confirmed the excision of the floxed Ercc1 allele in Lys2-Ercc1$^{F/-}$ (referred from now on as $Er1^{F/-}$) peritoneal macrophages (Fig. 1e), neutrophils and monocytes but not in neurons (Supplementary Fig. 1A). Confocal microscopy revealed the absence of ERCC1 expression in $Er1^{F/-}$ bone marrow-derived macrophages (BMDMs) and TEMs (Fig. 1f, Supplementary Fig. 1B) as well as in monocytes and neutrophils (Supplementary Fig. 1B). Together, these findings indicate the normative ERCC1 expression levels in $Er1^{F/-}$ tissues or cells other than the targeted cell populations i.e. monocytes, macrophages and neutrophils. Phosphorylated histone H2A.X ($\gamma$-H2A.X)-containing foci accumulate at sites of DNA breaks[31]. In line, the number of $\gamma$-H2A.X positive nuclei was significantly higher in the DNA repair-defective $Er1^{F/-}$ BMDMs (Fig. 1g) and TEMs (Fig. 1h) compared to Lys-Ercc1$^{F/+}$ control cells (referred from now on as $Er1^{F/+}$ or wild-type; wt.). We also find marked differences in the number of positively stained nuclei for FANCI involved in the repair of DNA ICLs[32],

RAD51 involved in the repair of DSBs by homologous recombination (HR)[33] and phosphorylated ATM, a central mediator of the DNA damage response (DDR; Fig. 1i for BMDMs and Fig. 1j for TEMs). Staining with caspase 3 revealed few, if any, apoptotic cells in $Er1^{F/-}$ TEMs (Fig. 1k). We also find no decrease in the total number of CD45(+) CD11b(+) macrophages during hematopoiesis (bone marrow) or at peripheral tissues i.e. blood, spleen and pancreas between $Er1^{F/-}$ and $Er1^{F/-}$ animals (Supplementary Fig. 1E–H). Staining with a lipophilic, biotin-linked Sudan Black B analogue (GL13; commercially available SenTraGor®[34]) in BMDMs revealed a uniform, >four-fold accumulation of lipofuscin (Fig. 1l). Likewise, senescence-associated (SA)-$\beta$-gal assay and western blotting for Lamin B1 revealed an increase in $\beta$-galactosidase expression in $Er1^{F/-}$ BMDMs (Supplementary Fig. 1C) and a mild but consistent senescence-associated loss of Lamin B1 expression in $Er1^{F/-}$ BMDMs, respectively (Supplementary Fig. 1D). Confocal microscopy studies with the HSP chaperone GRP78 known to be required for ER integrity[35], LC3$\beta$ that is stably associated with autophagosomal membranes[36], P62 a reporter of autophagic activity[37] and Gm130, a protein localized to the Golgi[38] revealed a dilated endoplasmic reticulum (ER; Fig. 2a), signs of autophagy with accumulation of autophagosomes (Fig. 2a; as indicated) and a dispersed and fragmented Golgi in $Er1^{F/-}$ macrophages (Fig. 2b). Western blotting further confirmed the increased GRP78, P62, and LC3 levels in $Er1^{F/-}$ macrophages (Fig. 2c). We also performed western blotting (Supplementary Fig. 2A) and/or confocal studies (Supplementary Fig. 2B) in $Er1^{F/+}$ BMDMs treated with nocodazole (known to trigger Golgi dispersal)[39], chloroquine (known to inhibit the degradation of autophagosomes in lysosomes)[40] and tunicamycin (known to trigger ER stress)[41] further confirming the validity of GRP78, P62, LC3, and GM130 biomarkers used to detect the cytoplasmic stress responses seen in $Er1^{F/-}$ macrophages (Supplementary Fig. 2B). To test whether persistent DNA damage triggers similar cytoplasmic stress responses, we exposed wt. macrophages to mitomycin C (MMC), a potent inducer of DNA ICLs. As with $Er1^{F/-}$ cells, treatment of wt. macrophages with MMC led to dilated ER lumen and autophagy (Fig. 2d; as shown) and to a dispersed Golgi throughout the cytoplasm (Fig. 2e). Exposure of wt. macrophages to lipopolysaccharide (LPS), a potent activator of monocytes and macrophages partially mimicked the MMC-driven autophagy and Golgi dispersal (Fig. 2f) but failed to instigate the dilation of ER. Inactivation of Ataxia-telangiectasia mutated (Atm) by exposing MMC-treated macrophages to KU-55933, an ATM inhibitor[42] or Ataxia telangiectasia and Rad3 related (ATR)/Cyclin dependent kinase inhibitor NU6027 (known to inhibit ATR kinase without interfering with irradiation-induced autophosphorylation of DNA-dependent protein kinase or ATM) reversed the Golgi dispersal after DNA damage (Fig. 2g). Likewise, inhibition of ATM and ATR dampened the increase in GRP78 and LC3 protein levels in MMC-treated macrophages (Fig. 2h) indicating that DNA damage is causal to ER dilation and autophagy; notably, the cytoplasmic alterations are reversible and require a functional DDR. Unlike in $Er1^{F/+}$ macrophages (Fig. 2i), transmission-electron microscopy in $Er1^{F/-}$ macrophages revealed the gradual accumulation of intracellular vesicles (Fig. 2j; left panel) often organized into larger vacuolar structures (Fig. 2j; right panel) and the appearance of cytoplasm-filled projections (Fig. 2k; left panel) that formed a convoluted network of pseudopodia-like structures (Fig. 2k; right panel) containing vesicles (Fig. 2l). These findings closely resemble the enhanced contractile activity observed in activated macrophages[43].

**The $Er1^{F/-}$ defect causes systemic metabolic alterations.** $Er1^{F/-}$ mice maintained a lower body weight (Fig. 3a) and a normal breeding efficiency over a period of 22 weeks. To investigate

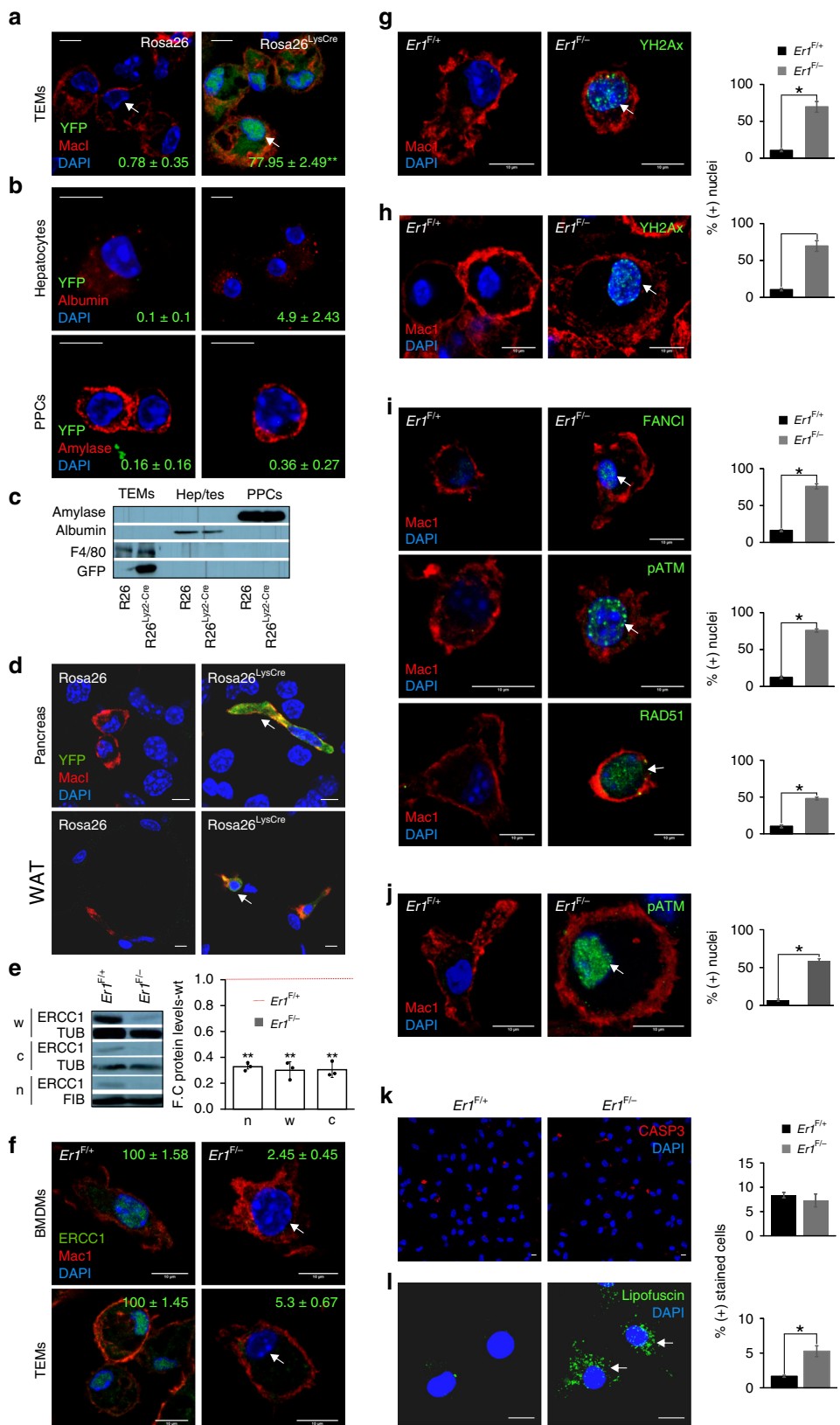

whether tissue-infiltrating $Er1^{F/-}$ macrophages trigger the onset of systemic metabolic complications in $Er1^{F/-}$ mice, we performed a glucose tolerance test (GTT) in 2-months-old $Er1^{F/-}$ and $Er1^{F/+}$ mice fed on a normal diet (ND) for a period of 2, 4 or 6 months. Animals were fasted for 16 h prior to administration of a single glucose dose (i.e. 200 mg/dL) and blood glucose levels

were measured at consecutive time intervals for a period of 3 h. After 2 months on ND, we find no difference in GTT between $Er1^{F/-}$ and $Er1^{F/+}$ animals (Fig. 3b; as shown). However, we find a progressive difference in GTT between $Er1^{F/-}$ and $Er1^{F/+}$ animals fed on ND for 4 or 6 months (Fig. 3b; as indicated) that was further pronounced when animals were fed for 4 months on

**Fig. 1 DNA damage accumulation in *Er1*^F/− macrophages. a** Lys-*Cre*-driven Rosa-YFP expression in thioglycolate-elicited peritoneal macrophages (TEMs; $n > 500$ cells counted per genotype). The numbers indicate the average percentage of GFP (+) cells ± SEM, **b** Lys-*Cre*-driven Rosa-YFP expression in hepatocytes and primary pancreatic cells (PPCs) shown by confocal microscopy ($n > 100$ cells counted per genotype; the numbers indicate the average percentage of GFP (+) cells ± SEM and **c** western blotting. **d** Immunofluorescence staining of Lys-*Cre*-driven Rosa-YFP expression in the *Er1*^F/− pancreas and the white adipose tissue (WAT) that are infiltrated with MAC1-positive macrophages (indicated by the arrowheads). **e** Western blotting of ERCC1 protein in whole-cell (w) cytoplasmic (c) and nuclear (n) extracts. Tubulin (TUB), and Fibrillarin (FIB) were used as loading controls (as indicated). The graph represents the fold change (F.C.) of ERCC1 protein levels in *Er1*^F/− samples compared to corresponding *Er1*^F/+ controls ($n = 3$). **f** Cell type-specific ablation of ERCC1 (indicated by the arrowhead) in bone marrow-derived (BMDMs) and TEMs expressing the macrophage-specific antigen MAC1. The numbers indicate the average percentage of ERCC1 (+) nuclei ± SEM in *Er1*^F/+ and *Er1*^F/− BMDMs and TEMs ($n > 150$ cells were counted per genotype). **g** Immunofluorescence detection of γ-H2AX in *Er1*^F/− and *Er1*^F/+ BMDMs and **h** TEMs. **i** Immunofluorescence detection of FANCI, pATM and RAD51 in *Er1*^F/− and *Er1*^F/+ BMDMs (in each case $n > 200$ cells were counted per genotype). **j** Immunofluorescence detection of pATM in *Er1*^F/− and *Er1*^F+ TEMs ($n > 150$ cells were counted per genotype). **k** Immunofluorescence detection of Caspase 3 (CASP3) ($n > 300$ cells were counted per genotype) and **l** GL13 (indicated by the arrowhead), commercially available SenTraGor®, in *Er1*^F/− and *Er1*^F+ BMDMs. Fluorescence intensity was calculated in $n > 50$ cells per genotype. Gray line is set at 5 μm scale, unless otherwise indicated. Error bars indicate S.E.M. among replicates ($n \geq 3$). Asterisk indicates the significance set at *p*-value: *≤0.05, **≤0.01 (two-tailed Student's *t*-test).

high-fat diet (Fig. 3c; as shown); unlike the *Er1*^F/− and *Er1*^F/+ animals fed on ND, *Er1*^F/− and *Er1*^F/+ animals fed on high-fat diet manifest no significant differences in body weight (Supplementary Fig. 2C), thereby, uncoupling *Er1*^F/− leanness from the noticeable differences in GTT and the mild, yet noticeable, hypoglycemia seen in 4- and 6-months-old *Er1*^F/− and *Er1*^F/+ animals (Fig. 3d). Further work revealed no difference in insulin serum levels (Fig. 3e) or insulin sensitivity (Fig. 3f) in the 8-months-old *Er1*^F/− animals and comparable levels of 2-Deoxy-D-glucose (2-DG) uptake in the liver and muscle protein extracts of overnight starved *Er1*^F/− and *Er1*^F/+ in response to insulin (Supplementary Fig. 2D). In line with enhanced glucose tolerance seen in *Er1*^F/− animals (Fig. 3b), we find that the 2-DG uptake is significantly higher in the tissues of non-insulin treated *Er1*^F/− mice when compared to *Er1*^F/+ corresponding controls (Supplementary Fig. 2D). Together these findings indicate that the enhanced glucose tolerance seen in *Er1*^F/− mice is mediated through alternative, insulin-independent mechanisms. Further work revealed an accumulation of glycogen (Fig. 3g) and the significant increase in the mRNA levels of glycogenin-1 (*Gyg-1*) and glycogen synthase (*Gys*) genes involved in the initiation of glycogen synthesis as well as the decrease in the mRNA levels of liver glycogen phosphorylase (*Pygl*) and glycogen synthase kinase 3 (*Gsk3*) involved in glycogen breakdown and the regulation of glycogen synthesis in the 6-months-old *Er1*^F/− livers (Fig. 3h). We find a lower epididymal fat and comparable body size (nasoanal length) in age-matched *Er1*^F/− and *Er1*^F/+ animals with lower deposition of triglycerides in *Er1*^F/− livers compared to age-matched *Er1*^F/+ control animals (Supplementary Fig. 3A; Fig. 3i) that remained substantially lower when *Er1*^F/− animals were maintained on a long-term high-fat diet (Fig. 3i; as indicated). The latter is in agreement with previous observations supporting the notion that accumulation of glycogen in the liver reduces food intake and attenuates obesity in mice[44]. The lower body weight in *Er1*^F/− mice (Fig. 3a) and the recent finding that LyzM-*Cre* is expressed in the hypothalamus[45] known to regulate appetite[46] prompted us to test for changes in the daily food intake of *Er1*^F/− mice. We find no differences in the daily food consumption of *Er1*^F+ and *Er1*^F/− mice over a period of 14 days (Supplementary Fig. 3B). Moreover, western blotting and immunofluorescence studies revealed comparable ERCC1 protein levels and no detectable accumulation of DNA damage-associated γ-H2A.X foci in the 7-months-old *Er1*^F+ and *Er1*^F/− hypothalamic regions (Supplementary Fig. 3C, D; as indicated). In support, the great majority of YFP signal colocalized with MAC1 in Rosa YFP transgenic animals expressing the Lys2-*Cre* transgene with detection of YFP signal in only a few e.g. 1–2 cells expressing the neuronal marker NeuN (Supplementary Fig. 3D).

*Er1*^F/− macrophages instigate a chronic inflammatory response. Histological examination in the 8-months-old *Er1*^F/− animals revealed the infiltration of monocytes and/or lymphocytes in perirenal fat, the kidney (Fig. 4a and Supplementary Fig. 3E), the lung and the liver (Fig. 4b and Supplementary Fig. 3F) and the presence of lipofuscin-accumulating macrophages in the white adipose tissue (Fig. 4c). Confocal studies revealed the infiltration of CD45+ hematopoietic cells and the accumulation of MAC1+ macrophages in the 8-months-old *Er1*^F/− livers (Fig. 4d), pancreata (Fig. 4e) and the white adipose tissue (Fig. 4f). These data and the previously documented higher frequencies of CD45+CD11b+ cells in all tissues tested (Supplementary Fig. 1E–H) along with the more differentiated/activated status of *Er1*^F/− monocytes/macrophages (as evidenced by their higher volume; Supplementary Fig. 3G) further support the presence of ongoing systemic inflammation. Flow cytometry (FACS) in the 8-months-old *Er1*^F/− and *Er1*^F/+ animals and in *Er1*^F/+ animals fed on a high fat diet (*Er1*^F/+ HFD) (Supplementary Fig. 4A) revealed a comparable increase in the infiltration of F4/80 (+) cells in the epididymal fat of *Er1*^F/− and *Er1*^F/+ HFD mice (HFD; Supplementary Fig. 4A); the latter animal group also developed insulin resistance as evidenced by the ITT test (Supplementary Fig. 4C). Unlike in *Er1*^F/+ HFD animals, FACS analysis with markers CD11c for M1[47] and CD206 for M2[48] revealed a potent increase in the M2 population of macrophages in *Er1*^F/− animals (Supplementary Fig. 4D). This finding indicates that the ERCC1 defect triggers the accumulation of macrophages that are polarized in a direction distinct to that seen in animal models associated with HFD and insulin resistance. In parallel, we find the increased expression of PECAM-1 involved in leukocyte transmigration and angiogenesis[49], ICAM-1 known to play a role in inflammatory and immune responses[50] and VCAM-1 involved in the adhesion of monocytes to vascular endothelium[51] (Fig. 4d–f). RNA-Seq profiling of *Er1*^F/− and *Er1*^F/+ macrophages revealed 1756 differentially expressed genes [meta-FDR ≤ 0.005, fold change ≥ ±1.5, 1090 upregulated genes; 665 downregulated genes; Fig. 4g; Supplementary data 1]. Within this gene set, the gene ontology (GO)-classified biological processes associated with innate immune or GTPase-mediated signaling, membrane-bound projection and pathways associated with endosomal/vascular and ER-phagosome interactions and senescence in *Er1*^F/− macrophages have a significantly disproportionate number of responsive genes relative to those mapped in the murine genome (false detection rate ≤0.05; Fig. 4h–j). Quantitative PCR for a set of pro-inflammatory cytokine and chemokine genes (Fig. 4k) and our previous findings (Figs. 2a–i and 1l, Supplementary Fig. 1C, D), confirmed the validity of gene expression changes.

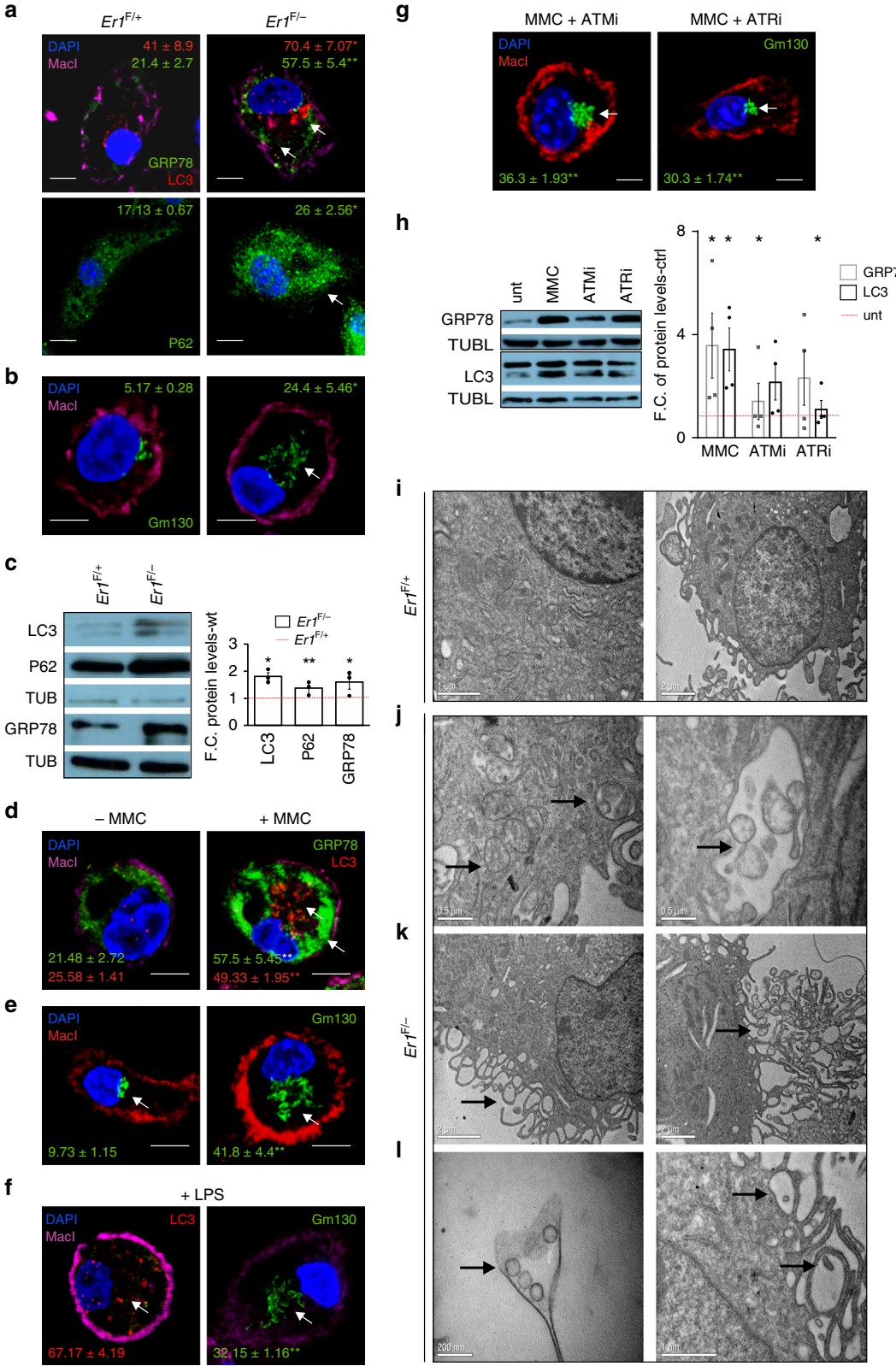

DNA damage triggers the release of extracellular vesicles. The increased tolerance of $Er1^{F/-}$ animals to glucose challenge and the activation of innate immune signaling in peripheral tissues prompted us to investigate whether $Er1^{F/-}$ macrophages acquire a secretory phenotype that exerts systemic, metabolic and pro-inflammatory stimuli in $Er1^{F/-}$ animals. To do so, we first employed a high-throughput mass spectrometry approach in isolated $Er1^{F/-}$ and $Er1^{F/+}$ macrophage media (Fig. 5a). This strategy led to the identification of 329 proteins with 211 proteins (64%) being shared by two independent measurements under stringent selection criteria (Fig. 5b and Supplementary data 2 and Methods). At the confidence interval used ($p$-value: $5 \times 10^{-11}$), we find that proteins associated with the presence of (membrane-bound) extracellular vesicles (EVs) and membrane-bound

**Fig. 2 Abrogation of ERCC1 triggers cytoplasmic stress responses in $Er1^{F/-}$ macrophages. a** Immunofluorescence detection of GRP78 ($n > 250$ cells counted per genotype) marking the dilation of ER, LC3 ($n > 40$ cells counted per genotype) for autophagy, P62 for autophagic activity ($n > 15$ optical fields per genotype, ~150 cells per field) and **b** Gm130 for Golgi dispersal (~500 cells per genotype; arrowhead) in $Er1^{F/-}$ and $Er1^{F/+}$ BMDMs. For GRP78 and LC3, the colored numbers indicate the average percentage of (+) stained cells ± SEM for the indicated, color-matched protein. For p62, the colored numbers indicate the average mean fluorescence intensity ± SEM of p62 signal. For Gm130, the green-colored numbers indicate the average percentage of (+) stained cells ± SEM showing Golgi dispersal. **c** Western blot levels of GRP78, P62, LC3, and Tubulin in $Er1^{F/-}$ and $Er1^{F+}$ BMDMs. The graph shows the fold change of indicated protein levels in $Er1^{F/-}$ BMDMs compared to $Er1^{F+}$ corresponding controls ($n = 3$ per group). **d** Immunofluorescence detection of GRP78 (for ER stress) ($n > 500$ cells counted per genotype), LC3 (for autophagy) ($n > 750$ cells counted per genotype) and **e** Gm130 (for Golgi dispersal) in MMC-treated BMDMs (n > 500 cells counted per genotype). Numbers indicate the average percentage of (+) stained cells ± SEM for the indicated, color-matched protein. For Gm130, the green-colored numbers indicate the average percentage of (+) stained cells ± SEM showing Golgi dispersal. **f** Immunofluorescence detection of LC3 (for autophagy) (~250 cells counted per genotype) and Gm130 (for Golgi dispersal) (~950 cells counted per genotype) in LPS-stimulated BMDMs. **g** Immunofluorescence detection of Gm130 (for Golgi dispersal) in MMC-treated and control BMDMs exposed to ATM (ATMi) (~200 cells counted) or ATR (ATRi) (500 cells counted) inhibitor (as indicated). The green-colored numbers indicate the average percentage of (+) stained cells ± SEM showing Golgi dispersal. **h** Western blot levels of GRP78 and LC3 in MMC-treated and control (ctrl) macrophages exposed to ATM (ATMi) or ATR (ATRi) inhibitor (as indicated; Tubl.: tubulin, unt: untreated). The graph represents the fold change in indicated protein levels in MMC-treated macrophages exposed to ATM (ATMi) or ATR (ATRi) inhibitor compared to corresponding controls ($n = 4$ per group) (**i–j**). Representative transmission electron micrographs of $Er1^{F+}$ (**i**) and $Er1^{F/-}$ (**j–l**) BMDMs. Arrowheads depict the presence of intracellular vesicles (**j** left panel), organized in larger vacuolar structures (**j** right panel), the appearance of cytoplasm-filled projections (**k** left panel), the convoluted network of pseudopodia-like structures (**k** right panel) containing vesicles (**l** left panel) and pseudopodia-associated extracellular vesicles (**l** right panel). Scale bars are shown separately for each micrograph. The significance set at $p$-value: *$\leq 0.05$, **$\leq 0.01$ (two-tailed Student's $t$-test). Gray line is set at 5 μm scale.

---

organelles are significantly over-represented (Fig. 5c) and present with a significantly higher number of known protein interactions (i.e. 178 interactions) than expected by chance (i.e. 109 interactions; $P \leq 8.23 \times 10^{23}$; Fig. 5d) indicating a functionally relevant and interconnected protein network. The most over-represented protein complex involved several RAB members of the RAS superfamily of GTPases (Fig. 5e) known to be associated, among others, with intracellular vesicle transport, the biogenesis and release of exosomes and the trafficking of glucose transporters to plasma membrane[52–60].

To validate the in vivo relevance of these findings, we isolated intact EVs from $Er1^{F/-}$ and $Er1^{F/+}$ sera. Western blot analysis confirmed the enrichment of vesicle-associated protein marker CD9, that is involved in the biogenesis, targeting and function of EVs[61] and ALIX known to be associated with the endosomal sorting complex required for transport[62] along with two members of the Ras superfamily of GTPases i.e. RAB10 and RAC1 (Fig. 5f, Supplementary Fig. 5A). Electron microscopy revealed the presence of EVs in the media of $Er1^{F/-}$ macrophages with a size ranging between 30 and 80 nm; the latter corresponds to that known for exosomes (40–100 nm diameter)[62], although the presence of larger (>200 nm) EVs cannot be excluded[63] (Fig. 5g). To test whether DNA damage is the primary instigator of EV secretion in $Er1^{F/-}$ macrophages, we exposed $Er1^{F/+}$ macrophages to the genotoxin MMC. In line with $Er1^{F/-}$ sera or the EV fraction of $Er1^{F/-}$ macrophage media (Fig. 5f–h, Supplementary Fig. 5A), we find a substantial enrichment for EV-associated protein markers CD9 and Alix and an accumulation of Ras GTPases RAB10, RAC2 and RAC1 in the EV fraction isolated from the media of MMC-treated cells compared to those of untreated cells (Fig. 5i). Further work revealed a similar accumulation of CD9 and Alix in the EV fraction of $Er1^{F/-}$ monocytes and neutrophils and a mild to negligible accumulation in the EV fraction of $Ercc1^{-/-}$ (designated from now on as $Er1^{-/-}$) primary mouse embryonic fibroblasts (MEFs) or the adipocytes, respectively (Supplementary Fig. 5B). Inactivation of DDR by inhibiting ATM or ATR in MMC-treated macrophages dampened substantially the increase in CD9, ALIX and RAB10, RAC2 and RAC1 protein levels (Fig. 5i and Supplementary Fig. 5C). The DNA damage-associated EV secretion is further supported by the parallel increase of γ-H2A.X, CD9 and Alix protein levels in BMDM whole-cell extracts or in the EV fraction of $Er1^{+/+}$, $Er1^{+/-}$, and $Er1^{-/-}$ BMDM culture media (Supplementary Fig. 5D). As detection of low abundance proteins (<100 ng/ml) is typically

challenging with current mass spectrometry protocols, we also employed an ELISA-based immunoassay to quantify Interleukin (IL)-1, IL6, IL-8, Interferon-γ, monocyte chemotactic protein 1, and stromal-derived-factor 1 in animal sera and macrophage media. We find substantially higher IL6 and IL8 levels in $Er1^{F/-}$ sera and macrophage media compared to $Er1^{F/+}$ control samples (Fig. 5j); all other cytokines were either non-detectable or did not vary significantly among the sample groups tested. Primary pancreatic cells (PPCs) or the hepatocytes are relevant cell types for coupling growth stimuli with fine-tuning mechanisms involved in nutrient-sensing and glucose homeostasis. Confocal studies confirmed the accumulation of RAC1, RAB10, RAC2, and RHOA in the cytoplasm of $Er1^{F/-}$ PPCs and hepatocytes (Fig. 5k, l and Supplementary Fig. 5E). In agreement with our previous findings on macrophage media (Fig. 5h), we also find that TEMs accumulate the exosome marker CD9 along with RAC1, RAB10 and RAC2 in their cell membrane (Fig. 5m); unlike other Ras GTPases tested, we find that RAC2 also accumulates in the nuclei of $Er1^{F/-}$ hepatocytes and TEMs (Fig. 5l, m; as shown). To test the functional relevance of RAB10 and RAC1 in the biogenesis/secretion of EVs in $Er1^{F/-}$ macrophages, we transfected $Er1^{F/+}$ macrophages with GFP-tagged RAB10 and RAC1. In line with the pronounced accumulation of CD9 and Alix in the EV fraction $Er1^{F/-}$ macrophages, we detect the marked accumulation of EV-associated protein markers CD9 and Alix in GFP-tagged RAB10/RAC1-transfected macrophages (Supplementary Fig. 6A). In parallel we treated $Er1^{F/-}$ macrophages with the RAC1 inhibitor NSC 23766 for 16 h[64]. We find that the inhibition of RAC1 activity in $Er1^{F/-}$ macrophages leads to a decrease in EV secretion, as evidenced by the decrease in CD9 protein levels (Supplementary Fig. 6B). Next, we performed live confocal imaging in $Er1^{F/-}$ and $Er1^{F/+}$ macrophages transiently expressing CD9-GFP. Unlike the sedentary appearance of $Er1^{F/+}$ macrophages (Supplementary Video File 1 and 2), we evidenced the gradual formation of a new vesicle-like structure in the cytoplasm (Supplementary Fig. 6C; white-colored arrow; Supplementary Video File 3), a fusion event of a cytoplasmic vesicle-like structure with the plasma membrane (Supplementary Fig. 6D; red-colored arrow, Supplementary Video File 3) and the presence of newly emerging pseudopodia in the membrane of $Er1^{F/-}$ macrophages (Supplementary Fig. 6E and Supplementary Video File 4).

$Er1^{F/-}$ EVs promote the glucose uptake in recipient cells. To test that $Er1^{F/-}$ macrophage-derived EVs ($Er1^{F/-}$ EVs) are delivered successfully to recipient cells, EVs were labeled with a

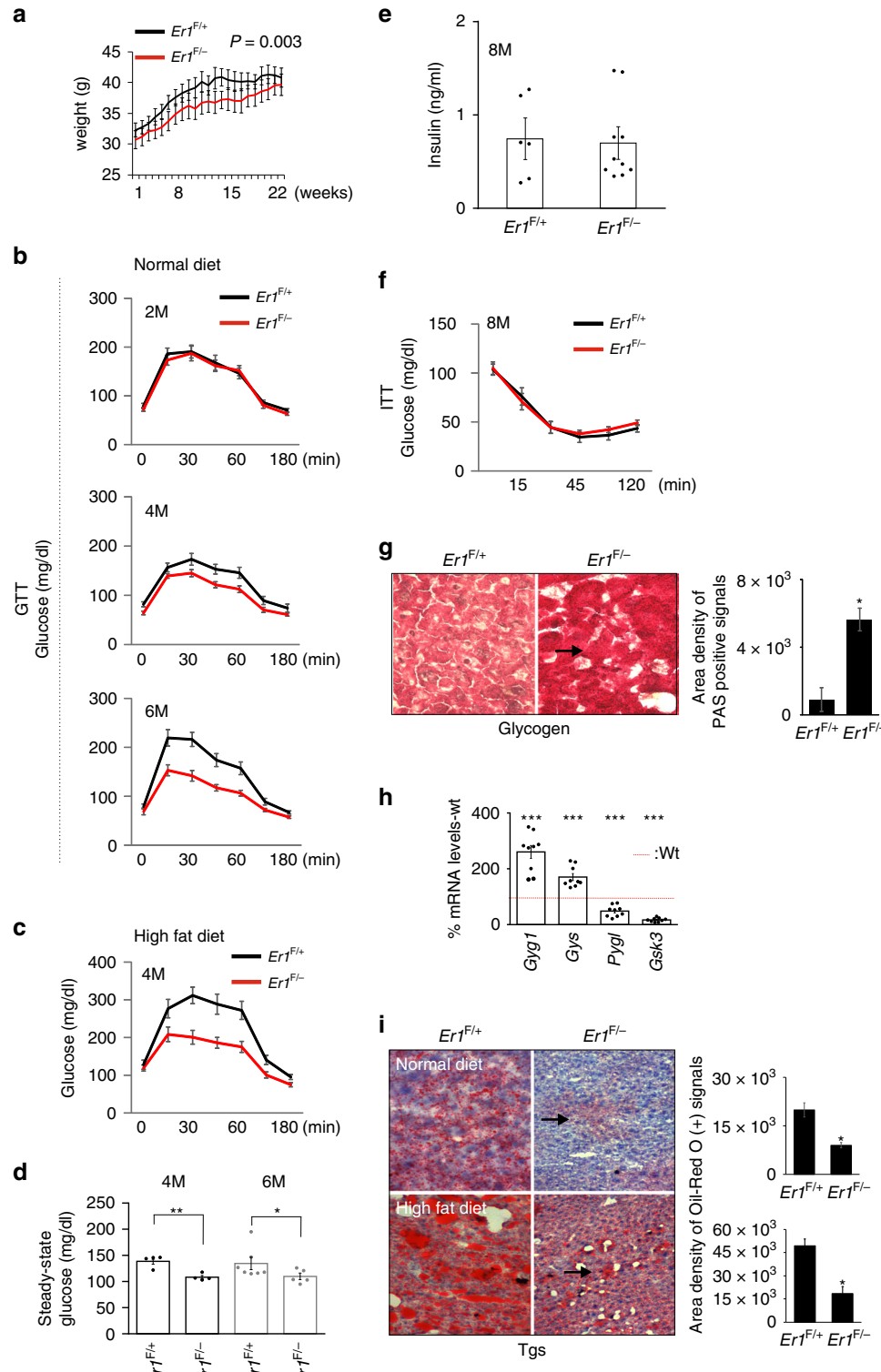

**Fig. 3 The ERCC-XPF defect in macrophages triggers metabolic changes in Er1$^{F/−}$ mice. a** Weights curves of 2-months-old Er1$^{F/−}$ and Er1$^{F/+}$ animals (n = 8) over a period of 22 weeks. **b** Glucose tolerance test (GTT) graphs of 2-months-old Er1$^{F/−}$ and Er1$^{F/+}$ mice fed on a normal diet for a period of 2-, 4-, and 6-months (M), as indicated. **c** GTT graphs of 2-months-old Er1$^{F/−}$ and Er1$^{F+}$ mice (n = 10) fed on a high-fat diet for a period of 4 months (M). **d** Steady-state glucose serum levels of 4- and 6-months (M) old Er1$^{F/−}$ and Er1$^{F/+}$ mice after 2 h of fasting (n = 8). **e** Insulin serum levels of 8-months (M) old Er1$^{F/−}$ (n = 10) and Er1$^{F/+}$ (n = 7) mice (**f**) Insulin tolerance test (ITT) graphs of 8-months (M) old Er1$^{F/−}$ and Er1$^{F/+}$ mice (n = 8). **g** Representative periodic acid–Schiff (PAS) staining and quantification (3 optical fields per animal) of glycogen in Er1$^{F/+}$ and Er1$^{F/−}$ livers (n = 4 animals per genotype). **h** Gyg1, Gys, Pygl, and Gsk3 mRNA levels in Er1$^{F/+}$ (red dotted line) and Er1$^{F/−}$ livers. **i** Representative Red-oil staining and quantification (three optical fields per animal) of triglycerides in the liver of Er1$^{F+}$ and Er1$^{F/−}$ mice (n = 3) fed on normal or high fat diet (as indicated); arrowhead indicates the decrease in fat deposition in the livers of Er1$^{F/−}$ animals fed on normal or high fat diet (as indicated). Error bars indicate S.E.M. among replicates (n ≥ 3). Asterisk indicates the significance set at p-value: *≤0.05, **≤0.01 (two-tailed Student's t-test). Gray line is set at 5 μm scale.

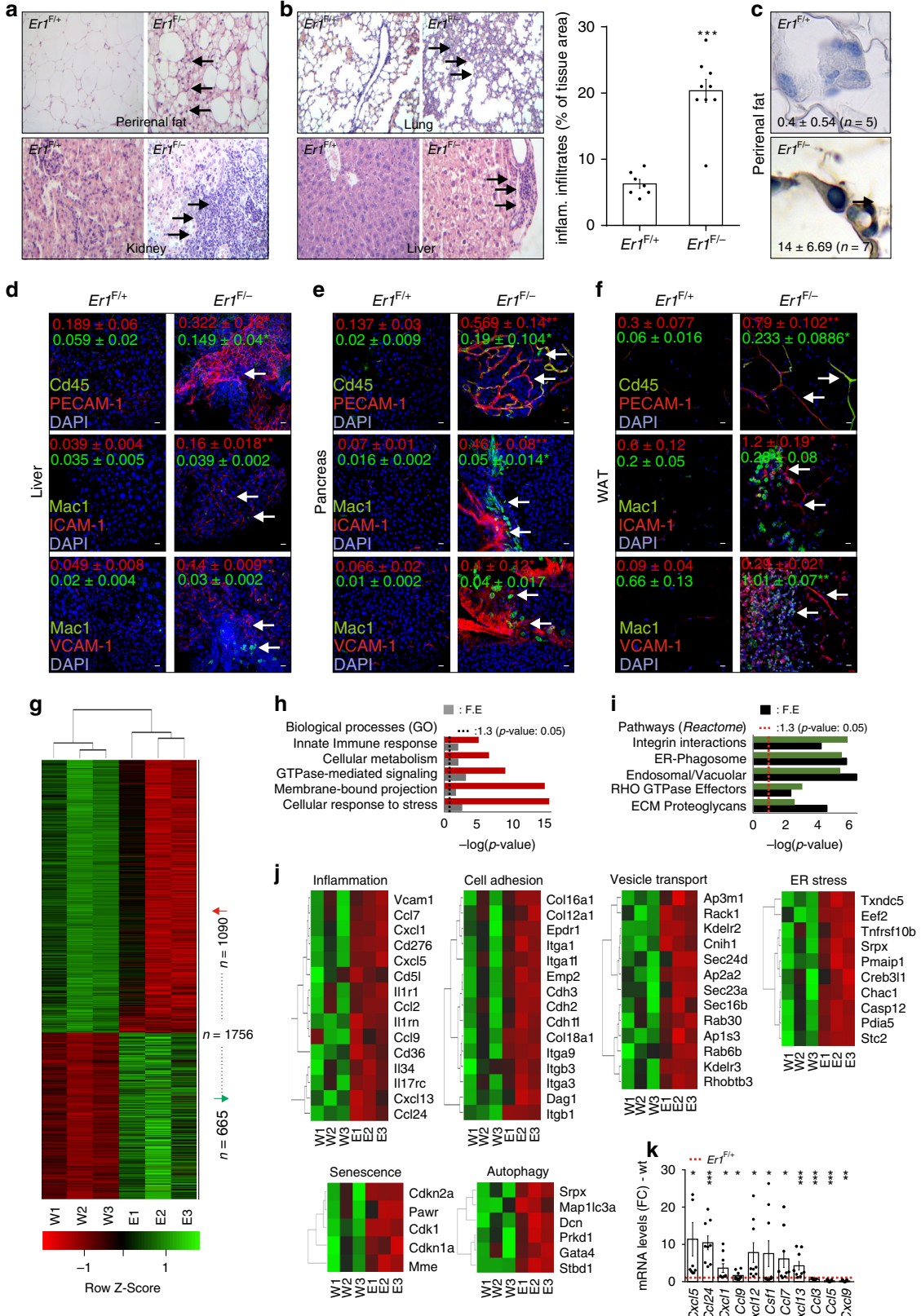

lipophilic green fluorescent dye PKH67 and were injected in the media of PPCs. Using this approach, we detect the presence of green-fluorescent, $Er1^{F/-}$ EVs in the cytoplasm of targeted PPCs (Fig. 6a). To further confirm that macrophage-derived $Er1^{F/-}$ EVs also deliver their cargo to recipient PPCs, we transfected $Er1^{F/-}$ macrophages with GFP-tagged RAB10 or RAC1

(Supplementary Fig. 7A). Following the exposure of PPCs to $Er1^{F/-}$ EVs carrying the GFP-tagged RAB10 and RAC1, we were able to detect both proteins in the cytoplasm of EV-recipient PPCs (Fig. 6b, c). Because several members of the RAB, RAS, and RHO family of small GTPases are known to regulate glucose uptake and are involved in the trafficking of glucose transporters

**Fig. 4 Systemic inflammation and gene expression changes in *Er1*$^{F/-}$ mice. a** Infiltration of foamy cells (macrophages) in a region of lipogranuloma in *Er1*$^{F/-}$ perirenal fat (as indicated) and inflammatory infiltration of lymphocytes and monocytes in *Er1*$^{F/-}$ kidneys. **b** Inflammatory infiltration of lymphocytes and monocytes in *Er1*$^{F/-}$ lungs and livers. See also Supplementary Fig. 3E, F for magnified inlays. The graph indicates the percentage of inflammatory (inflam.) infiltrates per tissue area in *Er1*$^{F/-}$ ($n = 9$) and *Er1*$^{F/+}$ mice ($n = 7$). **c** Detection of GL13 (+) macrophages in perirenal fat of *Er1*$^{F/-}$ ($n = 5$) and *Er1*$^{F/+}$ ($n = 7$) mice. **d** Immunofluorescence detection of PECAM-1, ICAM-1 and VCAM-1 along with CD45 and MAC1 (shown by the arrows) in the liver ($n = 4$; 4–6 optical fields per animal). **e** pancreas ($n = 3$; 3 optical fields per animal) and **f** the white adipose tissue (WAT) ($n = 3$; two optical fields per animal) of *Er1*$^{F/-}$ and *Er1*$^{F+}$ mice indicating the expression of cell adhesion molecules and the presence of monocytic/lymphocytic infiltrates in *Er1*$^{F/-}$ tissues. Colored numbers indicate the average mean fluorescence intensity ± SEM for the color-matched protein (as indicated). **g** Heat-map representation of significant gene expression changes ($n = 1756$ genes) in *Er1*$^{F/-}$ BMDMs compared to corresponding control cells. **h** Over-represented GO biological processes and **i** pathways (Reactome) of *Er1*$^{F/-}$ BMDMs compared to corresponding control cells; $p$: −log of $p$-value which is calculated by Fisher's exact test right-tailed, R: ratio of number of genes in the indicated pathway divided by the total number of genes that make up that pathway. **j** Heat-map representation of gene expression changes associated with significantly over-represented biological processes in *Er1*$^{F/-}$ BMDMs compared to corresponding control cells (as indicated). **k** Interleukin and chemokine mRNA levels in *Er1*$^{F/-}$ compared to *Er1*$^{F/+}$ (red dotted line) BMDMs. Error bars indicate S.E.M. among replicates ($n \geq 3$). Asterisk indicates the significance set at $p$-value: *$\leq 0.05$, **$\leq 0.01$ and ***$\leq 0.005$ (two-tailed Student's $t$-test), "+": one-tailed-$t$-test. F.E.: fold enrichment, W: (*Er1*$^{F/+}$), E: *Er1*$^{F/-}$. See also supplementary data 1. Gray line is set at 10 µm scale.

to cell membrane[65–68], we reasoned that the release of *Er1*$^{F/-}$ EVs in animal sera and the macrophage media promotes the glucose uptake in EV-recipient cells. Indeed, treatment of PPCs or hepatocytes with *Er1*$^{F/-}$ EVs triggers the noticeable uptake of 2-NBDG, a fluorescent tracer used for monitoring glucose uptake into living cells (Fig. 6d). We find similar findings when PPCs are exposed to EVs derived from *Er1*$^{F/-}$ sera (ser. *Er1*$^{F/-}$ EVs; Fig. 6e) or from *Er1*$^{F/+}$ macrophages exposed to the genotoxic agent MMC (MMC EVs; Fig. 6f).

Because the plasma membrane is impermeable to large polar molecules, such as glucose, the cellular uptake of glucose is accomplished through specific transmembrane transporters. Glucose transporters GLUT1 and GLUT3 are expressed in most tissues and, unlike other family members, they do not rely on insulin for facilitated diffusion of glucose across cell membranes[69,70]. Although GLUT1 was undetectable 3 h after the exposure of wt. PPCs to *Er1*$^{F/-}$ or MMC-EVs (Supplementary Fig. 7B and 7C; upper panel), it's mRNA and protein levels gradually accumulate within 24 h in these cells (Fig. 6g, h, Supplementary Fig. 7D; lower panel, 7E). Likewise, exposure of PPCs to *Er1*$^{F/-}$ or MMC-EVs induces the cytoplasmic accumulation of phosphorylated (S226) GLUT1 known to regulate the physiological regulation of glucose transport[71] (Supplementary Fig. 7F). Instead, RAB10 rapidly accumulates in the cytoplasm of PPCs within 3 h of exposure of cells to *Er1*$^{F/-}$ or MMC-EVs (Supplementary Fig. 7G) and is detectable with GLUT1 for at least 24 h (Fig. 6i and Supplementary Fig. 7H; as indicated). Using a previously established method[72], we also confirm a mild but reproducible increase in oxygen consumption rate (OCR) in PPCs treated with MMC- *Er1*$^{F/-}$- or serum-derived *Er1*$^{F/-}$ EVs (32%, 74%, and 26%, respectively) compared to cells exposed only to EVs derived from untreated cells or *Er1*$^{F/+}$ sera (Fig. 6j). Western blotting (Fig. 6k and Supplementary Fig. 8A) and immunofluorescence studies further confirmed the accumulation of GLUT 1 and 3 in the 6-months-old *Er1*$^{F/-}$ pancreata (Fig. 6l) and livers (Supplementary Fig. 8B). Staining for PECAM-1 and GLUT3 in *Er1*$^{F/-}$ tissues revealed that, for at least GLUT3, its expression is not restricted to endothelial cells but extends to other cell types in the *Er1*$^{F/-}$ livers, the white adipose tissue and the pancreata (Supplementary Fig. 8C).

Glucose uptake activates innate immune signaling. High–glycemic index diets have been associated with the activation of inflammatory processes[73]. These data and the marked inflammation seen in *Er1*$^{F/-}$ tissues (Fig. 4a–f, Supplementary Figs. 1E–H, 3G and 4A–D) led us to investigate whether the EV-mediated increase in glucose uptake triggers similar pro-inflammatory responses. To test this, we exposed PPCs to i. MMC-treated or *Er1*$^{F/-}$ macrophage media (the media are rich in

IL6 and IL8) (Fig. 5j) that were devoid of EVs or to ii. *Er1*$^{F/-}$ or MMC EVs alone (that were devoid of the macrophage media) or iii. to MMC-treated or *Er1*$^{F/-}$ macrophage media (the media are rich in IL6 and IL8) containing the secreted EVs. We find that the pro-inflammatory iNOS factor accumulates in PPCs exposed to MMC-treated or *Er1*$^{F/-}$ macrophage media containing the EVs (Fig. 7a, b). Interestingly, neither the *Er1*$^{F/-}$ EVs nor the MMC EVs alone or the media that were devoid of EVs could trigger a similar effect in iNOS accumulation (Fig. 7a and Supplementary Fig. 8D). Likewise, NF-κB that translocates to the nucleus to induce the transcription of pro-inflammatory genes[74,75] accumulates in the nuclei of PPCs exposed to macrophage media containing the MMC (Fig. 7c) or *Er1*$^{F/-}$ (Supplementary Fig. 8E) EVs; for NF-κB, exposure of PPCs to the media or the EVs alone also triggered a noticeable, albeit to a lesser extent, nuclear translocation (Supplementary Fig. 8E). These data suggest that EV-mediated glucose uptake activates iNOS or NF-κB signaling in cells previously primed with pro-inflammatory signals (in this case, IL6 and IL8). To test whether high glucose levels could potentiate the response, PPCs already exposed to MMC-treated or *Er1*$^{F/-}$ macrophage media containing the EVs were cultured under low (5 mmol) or high (15 mmol) glucose concentration. We find that iNOS accumulation and NF-κB nuclear translocation is more profound in PPCs maintained at the highest (15 mmol) glucose concentration (Fig. 7e, Supplementary Fig. 9A–C). High glucose levels are known to activate the PI3K/AKT/mTOR signaling pathway that couples cellular activation to environmental cues[76]. In line, we find a substantial increase in the protein levels of phosphorylated Eukaryotic translation initiation factor 4E (eIF4E)-binding protein 1 (P4E-BP1), a translation repressor protein and a well-known target of rapamycin (mTOR) signaling pathway, in *Er1*$^{F/-}$ pancreata (Fig. 7f). Inhibition of mTOR by exposing MMC-treated macrophages to rapamycin abrogated the increase in P4E-BP1 protein levels and iNOS accumulation (Fig. 7g). Likewise, rapamycin revokes the nuclear translocation of NF-κB in PPCs treated with *Er1*$^{F/-}$ EVs but does not affect glucose uptake (Fig. 7h).

Exogenous delivery of *Er1*$^{F/-}$ exosomes triggers inflammation. Having established that macrophage-derived *Er1*$^{F/-}$ EVs promote the glucose uptake in recipient cells and the activation of innate immune responses, we sought to test the specificity and in vivo relevance of our findings. To do so, B6 C57BL/6 animals were injected intravenously every 24 h for a period of 10 days with EVs derived from either *Er1*$^{-/-}$ or wt. MEFs and *Er1*$^{F/+}$ or *Er1*$^{F/-}$ macrophages. Unlike in animals treated with *Er1*$^{-/-}$ or wt. MEF-derived EVs and with *Er1*$^{F/+}$ macrophage-derived EVs, intravenous injection of animals with *Er1*$^{F/-}$ macrophage-derived EVs, leads to the pronounced accumulation of GLUT1 and the

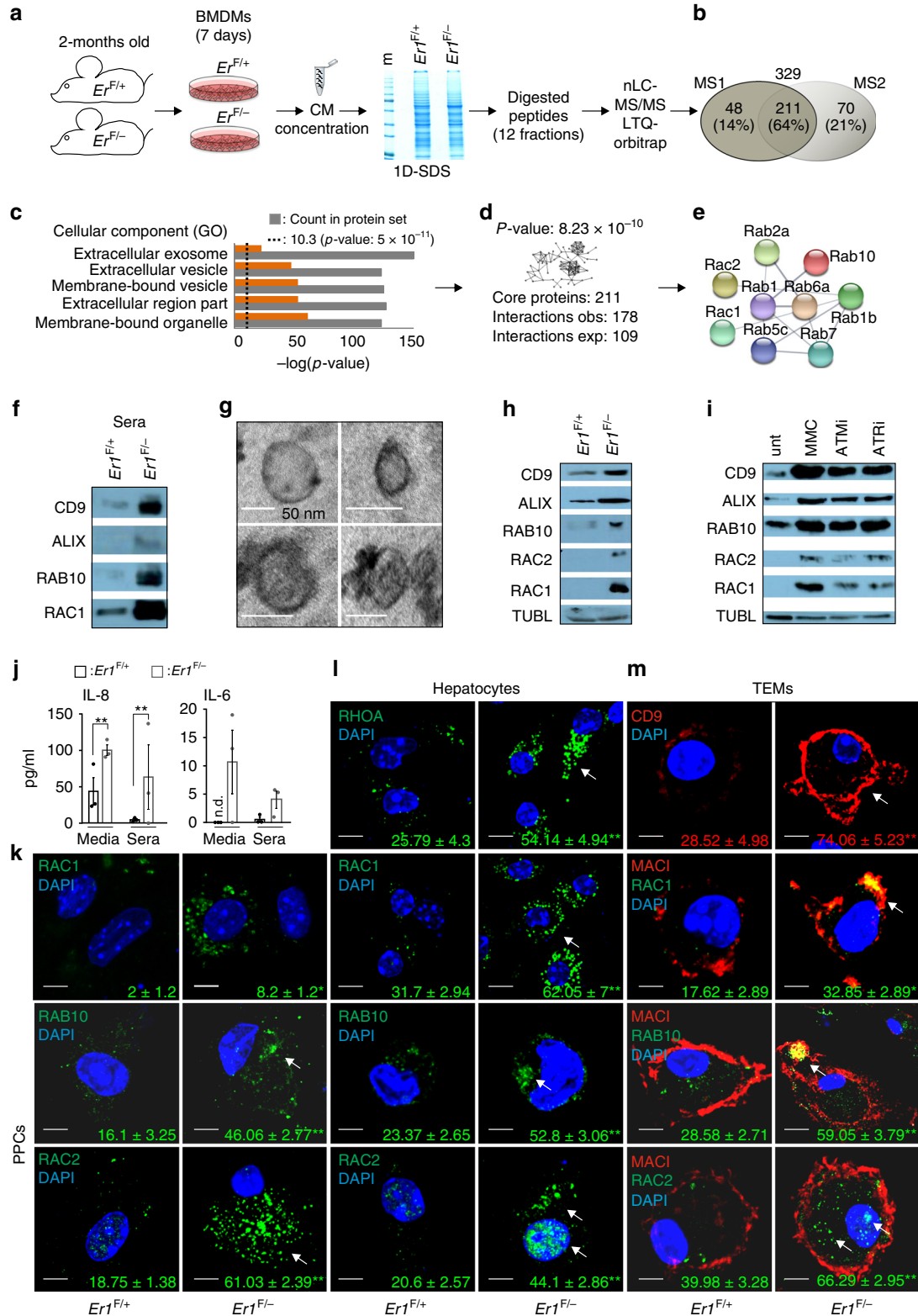

recruitment of macrophages in the liver of these animals (Fig. 8a, b). Western blotting further confirmed the accumulation of GLUT1 in the liver and muscle protein extracts of animals injected with $Er1^{F/-}$ macrophage-derived EVs but not with $Er1^{-/-}$ and wt. MEF-derived EVs or with $Er1^{F/+}$ macrophage-derived EVs (Fig. 8c, d). Importantly, mice injected with $Er1^{F/-}$ macrophage-derived EVs also manifest an enhanced glucose tolerance in GTT compared to corresponding control animals.

No difference in GTT is observed between animals treated with $Er1^{-/-}$ or wt. MEF-derived EVs (Fig. 8e; as indicated). In agreement, wt. PPCs treated with $Er1^{F/+}$ or $Er1^{F/-}$ EVs derived from either macrophages or monocytes and with EVs derived from $Er1^{-/-}$ or wt. adipocytes or MEFs leads to the accumulation of GLUT1 and iNOS (Supplementary Fig. 10A) and to the 2-NBDG uptake (Supplementary Fig. 10B) only in recipient cells exposed to $Er1^{F/-}$ macrophage-derived EVs.

**Fig. 5 DNA damage promotes the generation and secretion of extracellular vesicles (EVs) in Er1$^{F/-}$ macrophages. a** Schematic representation of the high-throughput MS analysis in Er1$^{F/-}$ compared to Er1$^{F/+}$ BMDMs media. **b** Venn's diagram of proteins identified in Er1$^{F/-}$ media from two independent biological replicates. **c** List of significantly over-represented GO terms associated with Cellular Component. **d** Number of observed (obs.) and expected (exp.) known protein interactions within the core 211 shared proteins set. **e** Schematic representation of the major protein complex identified in BMDM media. **f** Western blot analysis of CD9, ALIX, RAB10, and RAC1 proteins levels in the EV fraction of Er1$^{F/-}$ and Er1$^{F/+}$ sera ($n = 6$; see also Supplementary Fig. 5A; left panel). **g** Transmission electron microscopy of EVs marking the presence of exosomes with a size 30–80 nm in Er1$^{F/-}$ TEM media. **h** Western blot analysis of CD9, ALIX, RAB10, RAC2, and RAC1 proteins levels in Er1$^{F/-}$ compared to Er1$^{F/+}$ EV fraction of BMDM media ($n = 5$). A graph showing the fold change and statistical significance of the indicated protein levels is shown in Supplementary Fig. 5A; right panel. **i** Western blot analysis of CD9, ALIX, RAB10, RAC2, and RAC1 proteins levels in the EV fraction of media derived from the MMC-treated and control BMDMs exposed to ATM (ATMi) or ATR (ATRi) inhibitors (as indicated; $n = 3$). A graph showing the fold change and statistical significance of the indicated protein levels is shown in Supplementary Fig. 5C. **j** IL8 and IL6 protein levels in Er1$^{F/-}$ and Er1$^{F/+}$ sera and BMDM media (as indicated). **k** Immunofluorescence detection of RAC1 (~500 cells per genotype), RAB10 (~150 cells per genotype) and RAC2 (~150 cells per genotype) in Er1$^{F/-}$ and Er1$^{F/+}$ PPCs ($n > 400$ cells per genotype) (see also Supplementary Fig. 5e for RHOA), **l** hepatocytes ($n > 100$ cells per genotype) and **m** thioglycolate-elicited macrophages (TEMs) ($n > 500$ cells per genotype). Colored numbers indicate the average percentage of positively stained cells ± SEM for the indicated, color-matched protein. Error bars indicate S.E.M. among replicates ($n \geq 3$). Asterisk indicates the significance set at $p$-value: *$\leq 0.05$, **$\leq 0.01$ (two-tailed Student's $t$-test). (nd): not detected. Gray line is set at 5 μm scale.

## Discussion

How distinct cell types adapt their metabolic demands to counteract deleterious threats remains an intriguing question arguing for tissue-specific responses against irreparable DNA lesions. Our findings provide evidence that persistent DNA damage in circulating Er1$^{F/-}$ macrophages triggers the release of EVs that gradually surmount a systemic, glucose-based metabolic reprogramming leading to chronic inflammation in mice. The lack of apoptosis in Er1$^{F/-}$ macrophages indicates that the dilation of ER, Golgi dispersal and autophagy in these cells are not a byproduct of cell death. Instead, these processes occur quickly after DNA damage and are reversed when DDR is inhibited. Er1$^{F/-}$ macrophages form cytoplasmic projections filled in with EVs which are released in Er1$^{F/-}$ animal sera and the macrophage media and are rapidly secreted upon exposure of macrophages to DNA damage. The Er1$^{F/-}$ EV cargo is enriched with Ras GTPases involved in glucose transporter trafficking and vesicle-mediated transport[77] that accumulate in Er1$^{F/-}$ macrophages, PPCs, the hepatocytes and in wt. PPCs treated with Er1$^{F/-}$ or MMC EVs. Inhibition of ATM or ATR abrogates the release of MMC EVs highlighting the causal contribution of DNA damage signaling in this response. It remains to be seen whether the accumulation of lipofuscin and β-galactosidase along with the increase in the mRNA levels of p16INK4A, p21CIP1 and protein levels of IL-6 and IL-8 in Er1$^{F/-}$ macrophages reflect a recently proposed, reversible response of macrophages to physiological stimuli[78] or the premature onset of replicative or stress-induced senescence[79].

Er1$^{F/-}$ mice present with systemic inflammation and are hyper-tolerant to glucose challenge, independently, of insulin signaling. The response is progressive; it develops over the course of several months and is exacerbated when animals are fed on a long-term high-fat diet. We find that Er1$^{F/-}$ EVs tagged with the lipophilic green fluorescent dye PKH67 are successfully delivered to recipient PPCs and that macrophage-derived Er1$^{F/-}$ EVs carrying GFP-tagged RAB10 or RAC1 are detected in the cytoplasm of EV-recipient PPCs. Importantly, exposure of PPCs to Er1$^{F/-}$-, MMC- or serum-derived Er1$^{F/-}$ EVs leads to the noticeable uptake of glucose tracer 2-NBDG, the increase in GLUT1 protein and mRNA levels, the accumulation of RAB10 in the cytoplasm of PPCs and to higher OCR. Conversely, GLUT1 and GLUT3 known to facilitate cellular glucose uptake in an insulin-independent manner[80], accumulate in the Er1$^{F/-}$ livers and pancreata.

Hyperglycemia is acknowledged as a pro-inflammatory condition and lower glucose values are considered to be anti-inflammatory[81–83] indicating that the EV-mediated glucose uptake is causal to the inflammatory signals seen in Er1$^{F/-}$

tissues. However, exposure to high glucose concentrations does not cause inflammation unless cells are primed with an inflammatory stimulus[84,85]. Indeed, unlike in PPCs treated with only the IL6- and IL8-rich macrophage media or the EVs alone, exposure of PPCs to these media also containing the Er1$^{F/-}$ or MMC- EVs led to iNOS accumulation and to the nuclear translocation of NF-κB in EV-recipient cells; importantly, the response is more profound when cells are maintained at the highest glucose concentration. We find that the increase in glucose uptake activates mTOR signaling known to coordinate cell metabolism with environmental inputs, including nutrients[86]. In line, rapamycin, a potent mTOR inhibitor[87] abrogates 4E-BP1 phosphorylation, iNOS accumulation and the NF-κB nuclear translocation but has no effect on glucose uptake itself in PPCs, indicating that mTOR acts downstream of the glucose signaling pathway to activate pro-inflammatory responses. Recently, EVs have been used as natural nanocarriers for the systemic, in vivo delivery of biologically active cargo to recipient cells[88]. Importantly, we find that chronic exposure of animals to Er1$^{F/-}$ macrophage-derived EVs alone leads to GLUT1 accumulation, the recruitment of macrophages in parenchymal tissues and to enhanced glucose tolerance in wt. animals.

Previous data on the impact of defective NER in mammalian physiology[9,10,14,15,24,25] and the emerging role of exosomes in cell homeostasis[89–92], supports the notion that the DNA damage-driven metabolic adaptation represents a physiologic response that is both beneficial and detrimental for organismal survival. In the short run, activated macrophages could temporarily alert EV-recipient cells to rebuild their glucose reservoirs in order to defend themselves against foreign pathogens or threats. In the long run, however, the slow but steady buildup of irreparable DNA lesions e.g. in macrophages is expected to intensify EV secretion leading to the gradual onset of an exosome-based, metabolic reprogramming and pro-inflammatory signaling in mice. The latter would perpetuate a vicious cycle of persistent DDR signaling, increased glucose uptake and activated innate immune responses leading to the premature onset of chronic inflammation and tissue malfunction in NER progeroid syndromes (Fig. 8f). As DNA damage accumulates over time, a low glycemic diet may, therefore, be promising to delay age-related diseases[24,93].

## Methods

**Animal studies.** Ercc1$^{F/F}$ mice containing a floxed allele of the Ercc1 gene and Rosa26-YFP$^{st/st}$ mice were crossed with (Lys2)-Cre transgenic mice to obtain inactivation of the Ercc1 gene or expression of YFP in tissue-infiltrating macrophages, respectively. For insulin tolerance test (ITT), animals were fasted for 6 h

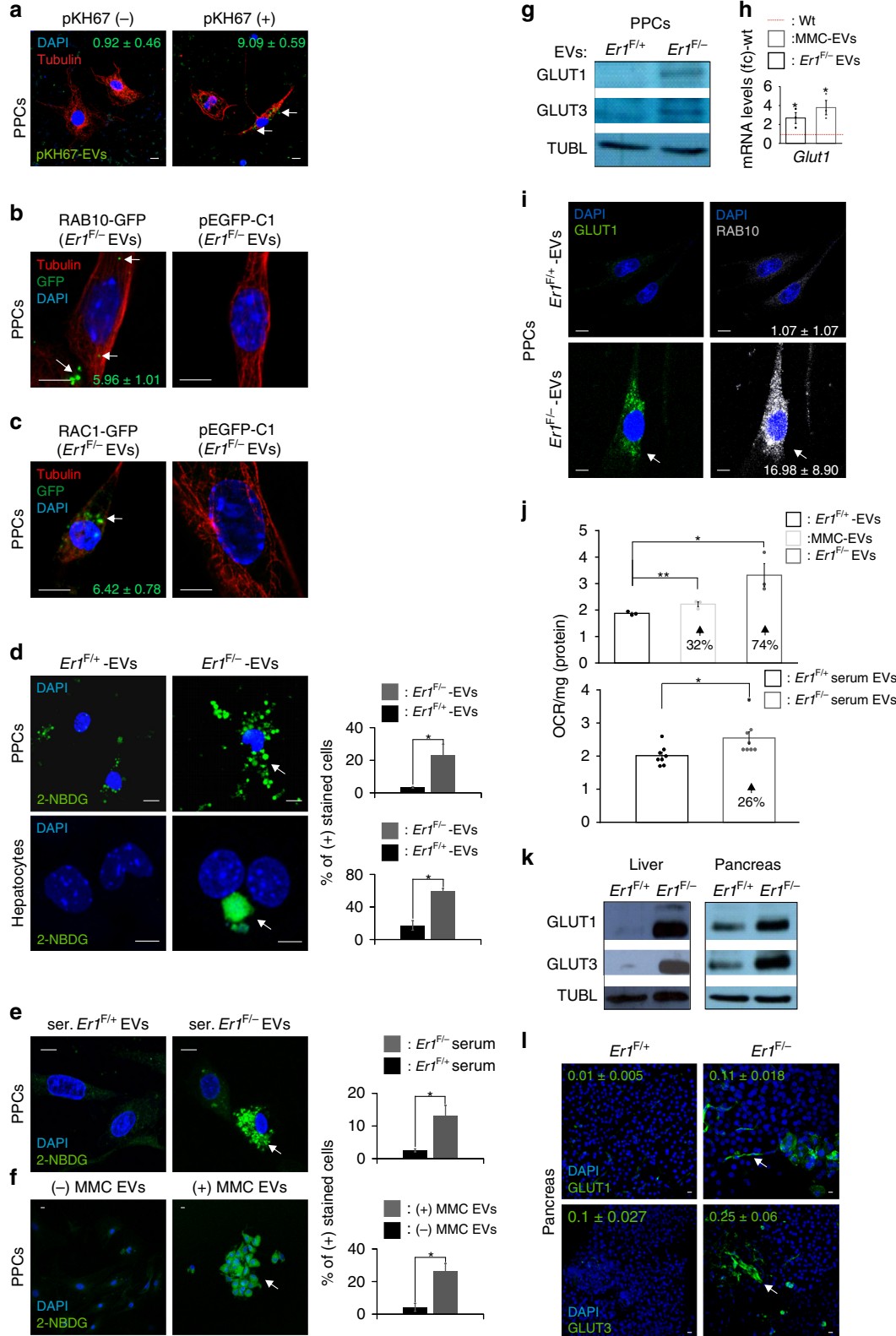

and were injected intraperitoneally with 0.75 Units/kg of body weight insulin (Humulin, Ely Lili). For glucose tolerance tests (GTT), mice were fasted for 16 h. and subsequently were injected intraperitoneally with 1 mg/gr of body weight 35% dextrose solution. Blood glucose levels were measured using CONTOUR® meter, at the indicated time points. To determine steady state glucose levels, 4- and 6-months-old animals were fasted for 2 h and glucose was determined. Serum insulin and triglyceride levels were measured with specialized kits (ALPCO and LabAssay triglyceride, Wako Chemicals, respectively). Mice were maintained in grouped cages in a temperature-controlled virus-free facility on a 12-h light/dark cycle and

fed either a high-fat diet (60% energy from fat, 20.3% carbohydrate, and 18.41% protein, 58Y1-58126, TestDiet) or a normal diet (Lactamin, Stockholm, Sweden). Mice had access to water ad libitum. Body weight was measured weekly. For food intake experiments, mice were kept individually in separate cages. Defined food quantity was added daily in each cage and food consumption was measured every 24 h. This work received ethical approval by and independent Animal Ethical Committee at the IMBB-FORTH. All relevant ethical guidelines for the work with animals were adhered to during this study. For the in vivo 2-DG uptake assay 6-months old male mice were used (three in each group). Animals were fasted O/N

**Fig. 6 $Er1^{F/-}$ EVs stimulate the glucose uptake in EV-recipient cells. a** Immunofluorescence detection of the pKH67-labelled EVs in PPCs. Numbers indicate the average percentage of positive cells ± SEM ($n > 800$ cells counted in three independent experiments). **b** Immunofluorescence detection of RAB10-GFP and **c** RAC1-GFP in PPCs treated with EVs from $Er1^{F/-}$ BMDMs transfected with RAB10-GFP or RAC1-GFP. Numbers indicate the average percentage of positively stained cells ± SEM for the indicated, color-matched protein. ($n > 700$ cells counted in three independent experiments). **d** Immunofluorescence detection of fluorescent tracer 2-NBDG for the monitoring of glucose uptake in PPCs (~2500 cells per genotype) and hepatocytes (~250 cells per genotype) exposed to $Er1^{F/-}$ and $Er1^{F/+}$-derived EVs (as indicated). **e** Immunofluorescence detection of fluorescent tracer 2-NBDG for the monitoring of glucose uptake in PPCs exposed to $Er1^{F/-}$ and $Er1^{F/+}$ EVs derived from animal sera (ser.; $n > 700$ cells per genotype). **f** Immunofluorescence detection of fluorescent tracer 2-NBDG for the monitoring of glucose uptake in PPCs exposed to MMC EVs (as indicated, $n > 2000$ cells per treatment). **g** Western blotting of GLUT1 and GLUT3 protein levels in PPCs exposed to $Er1^{F/-}$ and $Er1^{F/+}$-derived EVs ($n = 4$). A graph showing the fold change and statistical significance of GLUT1 and GLUT3 protein levels is shown in Supplementary Fig. 7E. **h** *Glut1* mRNA levels (in fold change; fc) in PPCs exposed to EVs derived from $Er1^{F/-}$ ($Er1^{F/-}$ EVs) or MMC-treated (MMC-EVs) macrophages compared to untreated wt. macrophages (red dotted line) for 24 h (see also Supplementary Fig. 7B–D). **i** Immunofluorescence detection of GLUT1 and RAB10 in PPCs exposed to $Er1^{F/-}$ and $Er1^{F+}$-derived EVs (see also Supplementary Fig. 7G, H). Numbers indicate the average percenatge of GLUT1, RAB10 double positive cells in each experimental condition ($n > 200$ cells counted in three independent experiments). **j** Oxygen consumption rate (OCR) of PPCs maintained in the presence of 15 mmol glucose that were exposed to $Er1^{F/+}$, $Er1^{F/-}$ or MMC-treated, macrophage-derived EVs (upper panel) and to $Er1^{F/+}$ or $Er1^{F/-}$ serum-derived EVs (lower panel); numbers indicate the % increase in OCR compared to corresponding controls. **k** Western blotting of GLUT1 and GLUT3 protein levels in $Er1^{F/-}$ and $Er1^{F/+}$ liver and pancreata ($n = 3$). A graph showing the fold change and statistical significance of GLUT1 and GLUT3 protein levels is shown in Supplementary Fig. 8A. **l** Immunofluorescence detection of GLUT1 and GLUT3 in the pancreas of $Er1^{F/-}$ and $Er1^{F/+}$ mice ($n = 3$, 2–3 optical fields per animal; see also Supplementary Fig. 8B). Numbers indicate the average mean fluorescence intensity. Grey line is set at 10 μm scale. Error bars indicate S.E.M. among replicates ($n \geq 3$). Asterisk indicates the significance set at $p$-value: *$\leq 0.05$, **$\leq 0.01$ (two-tailed Student's $t$-test). Gray line is set at 5 μm scale.

and 2-DG (200 μmol/kg) was injected into the tail vein 15 min after intraperitoneal insulin administration (1 Units/kg of body weight). Liver, soleus and rectus femoris muscle samples were collected 15 min after 2-DG injection. Tissues were digested with collagenase (2.5 mg/ml) and dispase II (2.4 units/ml) at 37 °C for 15 min. Collagenase was neutralized with 10%FBS and samples were further washed with 1x PBS/1%BSA. Homogenized tissues were passed through a 100 μM wire mesh. Cells were resuspended in 10 mM Tris-HCl pH 8.0 and disrupted with a microtip sonicator and heat treatment at 80 °C for 15 min. Protein extracts were measured with Bradford and equal amounts of protein extracts (200 μg) were used to determine intracellular 2-DG–6-phosphate levels with a 2-DG uptake measurement kit (Cosmo Bio Co., Ltd.) according to manufacturer instructions. For the exogenous delivery of EVs 4-weeks-old B6 mice were injected intravenously every 24 h for 10 days with EVs isolated from media of $10 \times 10^6$ cells ($Er1^{-/-}$ or wt. MEFs and $Er1^{F/+}$ and $Er1^{F/-}$ BMDMs) as described in the section EV isolation. On day 10 mice were starved O/N and GTT was performed as described above. Mice were sacrificed and liver and muscle tissues were isolated and analyzed with Western blotting and Immunofluorescence.

Electron microscopy. For transmission electron microscopy, primary macrophages were washed in PBS followed by fixation in 2% paraformaldehyde, 2% glutaraldehyde in 0.1 M sodium cacodylate buffer (pH 7.42) with 0.1% magnesium chloride and 0.05% calcium chloride. After washes with sodium cacodylate buffer, cells were fixed in 1% osmium tetroxide in sodium cacodylate buffer and samples were dehydrated in ethanol gradient. Samples were then treated with propylene oxide and embedded in Epon/Araldite resin mix. Ultrathin-sections (50–100 nm) were taken on a Leica LKB2088 ultramicrotome and were examined under JEM 100 C/JEOL/Japan Transmission Electron Microscope. Microphotographs were obtained with an ES500W Erlangshen camera and processed with the Digital Micrograph software (Gatan, Germany). For electron microscopy (EM) analysis of whole-mount exosome preparations, fixed EVs were deposited on EM grids and were further fixed with glutaraldehyde. Samples were first contrasted in a solution of uranyl oxalate and then contrasted and embedded in a mixture of 4% uranyl acetate and 2% methyl cellulose.

EV isolation and labelling. Exosomes were purified using the differential ultracentrifugation protocol[94]. Briefly culture medium was centrifuged sequentially at 300 g, (10 min), 2000 g (10 min), and 10000 g (30 min) to remove dead cells and cell debris. Extracellular vesicles were purified with the final step of ultracentrifugation at 100000 g for 2 h. For functional experiments, macrophage-derived EVs were purified five times the number of recipient cells. For PKH67 staining, EVs were incubated with PKH67 (500 mL 0.2 mM) for 5 min at room temperature. Labelled EVs were diluted in 500 mL 1% BSA, and then pelleted at 100,000 g, washed with 1 mL PBS to remove excess dye, re-suspended in 1 mL PBS and then pelleted at 100,000 g before final re-suspension. The same protocol was followed in the absence of EVs (dye alone) to ensure lack of fluorescence. For functional experiments, macrophage-derived EVs were purified five times the number of recipient cells and then added in the medium of PPCs or hepatocytes for 3 or 24 h.

Histology, Immunofluorescence and Immunoblot analysis. $Er1^{F/+}$ and $Er1^{F/-}$ livers were OCT-embedded, cryosectioned, fixed in 10% formalin, stained with oil red O or Periodic acid–Schiff (PAS), counterstained with Harris's hematoxylin and visualized with DAB chromogen (Sigma). For histological analysis of $Er1^{F/+}$ and $Er1^{F/-}$ tissues, samples were fixed in 4% formaldehyde, paraffin embedded, sectioned and stained with Harris's Hematoxylin and Sudan Black B (SBB)-Analogue (GL13). For immunofluorescence experiments of mouse tissues, minced tissues (liver, pancreas and hypothalamus), BMDMs, TEMs, PPCs and primary

hepatocytes were fixed in 4% formaldehyde, permeabilized with 0.5% Triton-X and blocked with 3% normal calf serum and 1% BSA. After overnight incubation with primary antibodies, secondary fluorescent antibodies were added and DAPI was used for nuclear counterstaining. Samples were imaged with SP8 confocal microscope (Leica). For SDS-page analysis, EVs were derived from $20–25 \times 106$ macrophages, whole-cell extracts and nuclear or cytoplasmic extracts were used. Rapid time-lapse imaging was performed using SP8 confocal microscope (Leica) with a stage-top incubation system creating a 37 °C environment. Z-stacks of 3–4 sections with two-channel detection were acquired every. For live cell experiments, in vitro differentiated $Er1^{F/+}$ and $Er1^{F/-}$ BMDMs were transfected with 5 μg of total plasmid DNA (pEGFP-CD9, see below). Transfected cells were seeded on a Mattek culture dish and imaged 24 h post transfection. Time-lapse stacks were analyzed with Fiji. At least 20 cells from each genotype were recorded in total in three independent replicates.

Primary cell cultures, transfection and cell assays. $Er1^{F/+}$ and $Er1^{F/-}$ BMDMs were differentiated from bone marrow precursors. Briefly, bone marrow cells were isolated from mouse femurs and tibias and cultured for 7 days in DMEM containing 10% FBS, 30% L929 conditioned media, 50 μg/ml streptomycin, 50 U/ml penicillin (Sigma) and 2 mM L glutamine (Gibco). For TEM isolation, mice were injected intraperitoneally with 2 ml 4% thioglycolate medium (Brewer). Three days later, mice were sacrificed and peritoneal cells were collected isolated by peritoneal lavage with 10 ml dMEM per mouse. Primary hepatocytes and PPCs were obtained from 20-days old mice. Briefly, liver and pancreas were excised, minced and incubated in 2 mg/ml collagenase type IV at 37 °C for 15 min. After centrifugation, cells were resuspended in DMEM containing 10% FBS, 50 μg/ml streptomycin, 50 U/ml penicillin (Sigma) and 2 mM L glutamine (Gibco). Before any experiment, primary hepatocytes were cultured overnight and PPCs for 4 days with medium replacement daily. For isolation of monocytes, marrow cells were cultured for five days in DMEM containing 10% FBS, 30% L929 conditioned media, 50 μg/ml streptomycin, 50 U/ml penicillin (Sigma) and 2 mM L glutamine (Gibco). Monocytes were harvested by collecting the non-adherent cells using an EDTA-free wash buffer. For neutrophil isolation, marrow cells were isolated from the cell suspension by density gradient centrifugation on Percoll. The Percoll density gradient was prepared in a 15 ml tube by layering 2 ml of 67 and 52% Percoll solution on top of 2 ml 75% Percoll solution. Cells were resuspended in 2 ml of PBS and loaded on top of the Percoll density gradient. Neutrophils were separated by centrifugation at 400 g for 30 min at 4 °C in a swinging bucket rotor. The cell band formed between the 75 and 67% layer was harvested, cells were diluted with PBS, washed, and diluted in standard medium. Monocytes and neutrophils were cultured for 15 h in low oxygen incubator prior to EV isolation. Primary MEFs and adipocytes were derived from $Ercc1^{-/-}$ animals. Primary MEFs were induced 2 days after confluency for adipocyte differentiation with standard medium supplemented with an adipogenic cocktail (0,5 mM IBMX, 1 μM dexamethasone, 10μgr/ml insulin)[15]. BL/6 or $Er1^{F+}$ BMDMs or $Er1^{F/-}$ were treated with 2.5 μg/μl mitomycin C (MMC; AppliChem) or tunicamycin (1μgr/ml), chloroquine (10μM), nocodazole (5 μM) and NSC23766 (50μM Rac1 inhibitor), for ~15 h in standard medium. For ATM or ATR kinase inhibitor assays (ATMi; ATRi), cells were incubated with 0.66 μM ATMi or ATRi (Millipore) for 1 h, prior to MMC addition. For plasmid transfections, approximately $1 \times 10^6$ differentiated BMDMs were transfected with 5 μg of total plasmid DNA using Amaxa mouse macrophage nucleofector kit according to the manufacturer's instructions (Lonza, VPA1009BMDMs were incubated for 24 h and their media were used for EV isolation. To generate the RAB10-GFP and RAC1-GFP and CD9-GFP, the cDNA encoding the whole open reading frame

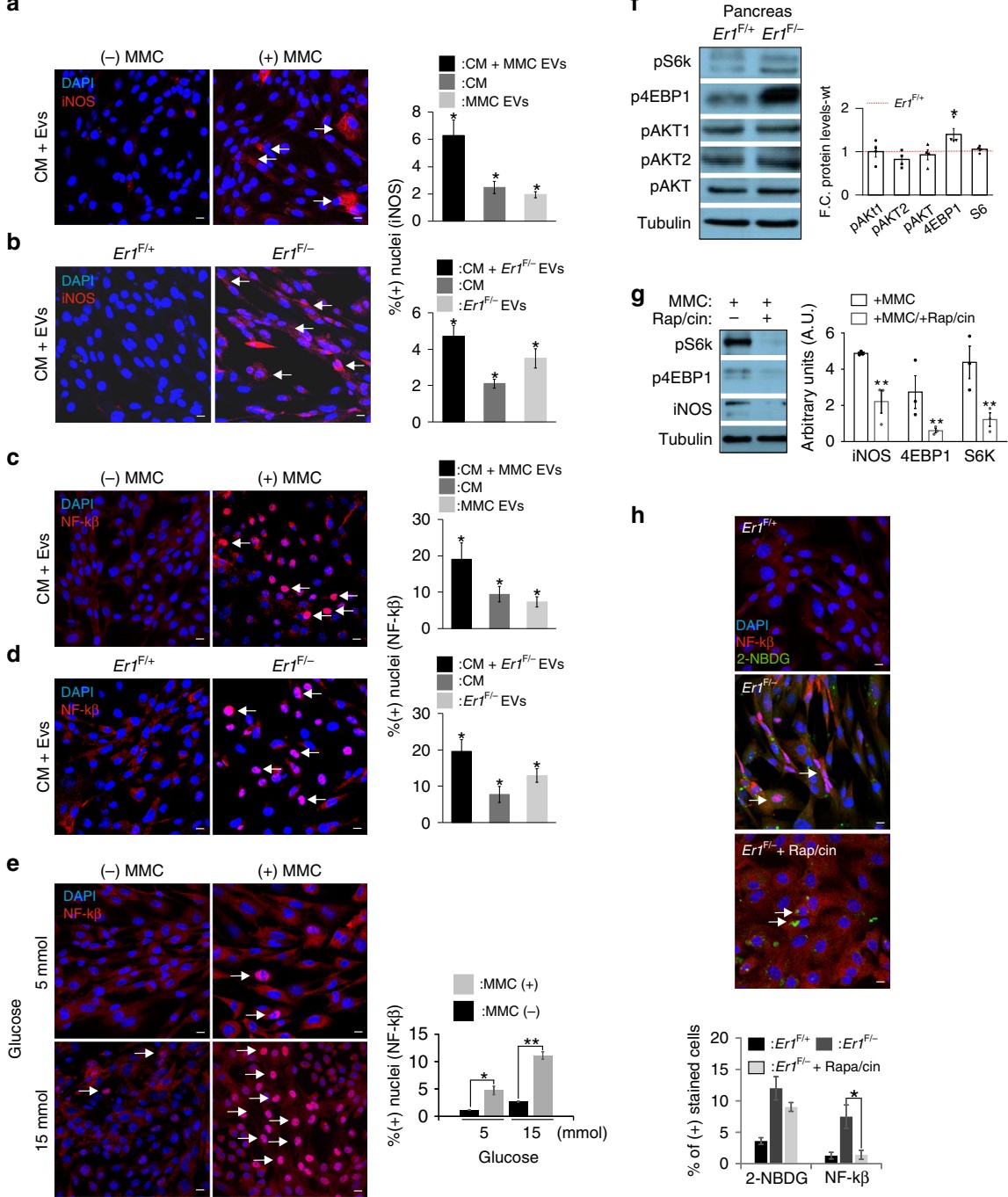

**Fig. 7 Glucose uptake activates mTOR and pro-inflammatory responses in EV-recipient cells. a** Immunofluorescence detection of iNOS accumulation (indicated by the arrowhead) in PPCs exposed to culture media (CM) and the EVs derived from MMC-treated and untreated control macrophages ($n = 3$, >600 cells per treatment) or **b** $Er1^{F/-}$ and $Er1^{F/+}$ macrophages ($n = 3$, >1200 cells/treatment) (see also Supplementary Fig. 8D). **c** Immunofluorescence detection of NF-kβ nuclear translocation (indicated by the arrowhead) in PPCs exposed to CM and the EVs derived from MMC-treated and untreated control macrophages or ($n = 3$, >700 cells per treatment). **d** $Er1^{F/-}$ and $Er1^{F/+}$ macrophages (see also Supplementary Fig. 8E). ($n = 3$, >500 cells per treatment). **e** Immunofluorescence detection of NF-kβ (shown by the arrowhead) in PPCs exposed to CM supplemented with EVs from MMC-treated and control macrophages upon low (5 mmol) or high (15 mmol) glucose concentration ($n = 4$, >750 cells per treatment; see also Supplementary Fig. 9A–C). **f** Western blotting of phosphorylated pS6K, phosphorylated p4EBP1, phosphorylated pAKT1, phosphorylated pAKT2 and phosphorylated pAKT protein levels in $Er1^{F/-}$ and $Er1^{F/+}$ pancreata ($n = 4$). The graph represents the fold change (F.C) of indicated protein levels in $Er1^{F/-}$ pancreata to wt. controls. **g** Western blotting of pS6K, p4EBP1 and iNOS protein levels in MMC-treated macrophages exposed to rapamycin ($n = 3$, Rap/cin; as indicated). The graph represents the fold change of protein levels in MMC-treated macrophages exposed to rapamycin compared to MMC-treated macrophage control (ctrl.) cells. **h** Immunofluorescence detection of NF-kβ and 2-NBDG in PPCs (indicated by the arrowheads) exposed to CM supplemented with EVs derived from $Er1^{F/-}$ and $Er1^{F/+}$ BMDMs in the presence or absence of rapamycin (Rap/cin). ($n = 6$, >1000 cells counted per treatment). The graph shows the percenatge of positively stained cells. Error bars indicate S.E.M. among replicates ($n \geq 3$). Asterisk indicates the significance set at p-value: *$\leq 0.05$, **$\leq 0.01$ (two-tailed Student's t-test). Gray line is set at 5 μm scale.

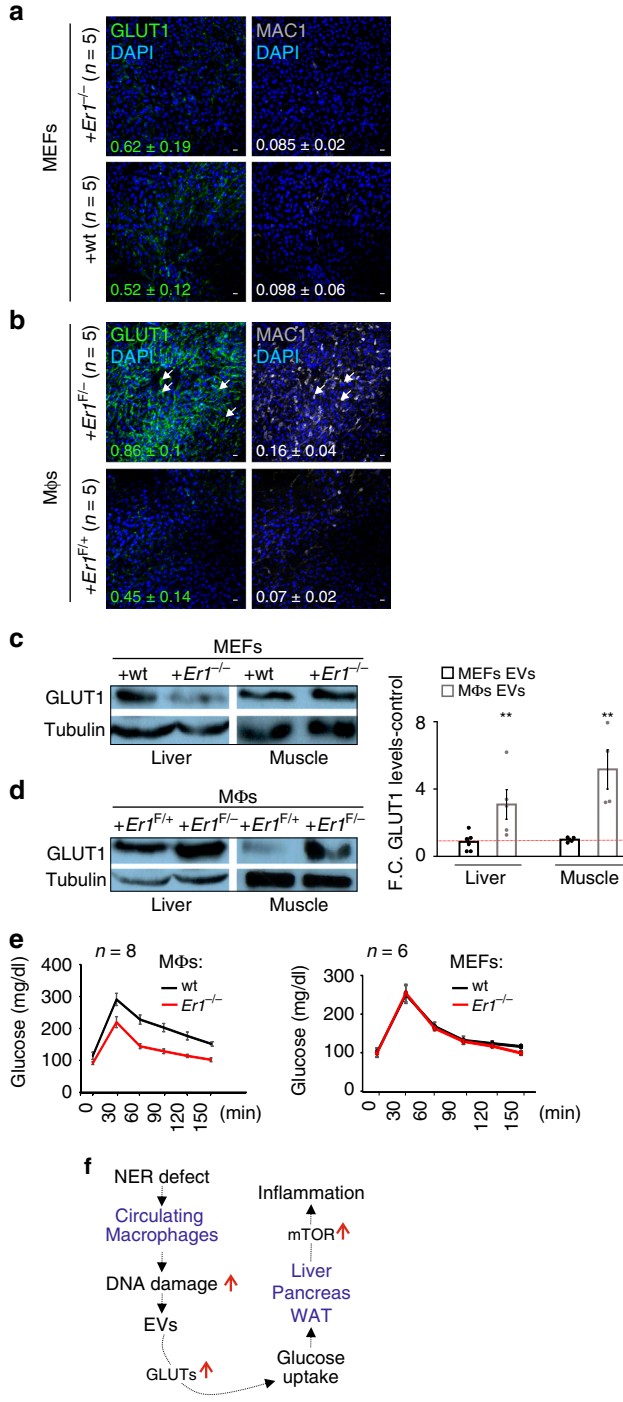

**Fig. 8 Exogenous delivery of *Er1*<sup>F/−</sup> exosomes promote GLUT1 accumulation, glucose uptake and inflammation in vivo. a** Immunofluorescence detection of GLUT1 in the liver of 4-weeks-old C57BL/6 animals intravenously injected with EVs derived from either *Er1*<sup>−/−</sup> or wt. MEFs and **b** *Er1*<sup>F/+</sup> or *Er1*<sup>F/−</sup> macrophages (as indicated). Numbers show average mean fluorescence intensity ± SEM, $n = 4$ animals per genotype, >3 optical fields/animal. **c** Western blotting of GLUT1 in the muscle and liver of C57BL/6 animals intravenously injected with EVs derived from either *Er1*<sup>−/−</sup> or wt. MEFs and **d** *Er1*<sup>F/+</sup> or *Er1*<sup>F/−</sup> macrophages (as indicated). The graph represents the fold change (F. C) of GLUT1 protein levels in the liver or muscle tissues of animals treated with *Er1*<sup>−/−</sup> MEF or *Er1*<sup>F/−</sup> macrophage-derived EVs as compared to corresponding tissues of animals treated with *Er1*<sup>+/+</sup> MEFs or *Er1*<sup>F/+</sup> macrophage-derived EVs. **e** Glucose tolerance test (GTT) graphs of C57BL/6 mice injected intraperitoneally every 24 h for a period of 10 days with EVs derived from either *Er1*<sup>−/−</sup> or wt. MEFs ($n = 6$) and *Er1*<sup>F/+</sup> or *Er1*<sup>F/−</sup> macrophages ($n = 8$) (as indicated). **f** The accumulation of irreparable DNA lesions in tissue-infiltrating macrophages activates the secretion of EVs in vivo and ex vivo. *Er1*<sup>F/−</sup> macrophage-derived EVs are targeted to recipient tissues and cells triggering the expression and translocation of insulin-independent glucose transporters GLUT1 and 3 in cell membrane. This leads to an increase in cellular glucose uptake in cells and greater tolerance to glucose challenge in higher cellular oxygen consumption rate and greater tolerance to glucose challenge in *Er1*<sup>F/−</sup> mice. In turn, high glucose levels activate pro-inflammatory responses in an mTOR-dependent manner leading to chronic inflammation in *Er1*<sup>F/−</sup> animals. Error bars indicate S.E.M. among replicates ($n$ is indicated in each panel). Asterisk indicates the significance set at $p$-value: *$\leq 0.05$, **$\leq 0.01$ (two-tailed Student's $t$-test). Gray line is set at 10 μm scale.

either experimental or control plates (3 h). Cells were washed in PBS, and the numbers of β-gal-positive cells (blue staining) in at least 350 cells were counted in random fields in each of the triplicate wells. For Sentragor staining cells were incubated with Sentragor reagent according to manufacturer instructions (Arriani pharmaceuticals, AR8850020). Briefly fixed differentiated BMDMS and TEMs were washed with 50% and 70% EtOH incubated with Sentragor reagent at RT, washed in EtOH and incubated with primary anti-biotin Ab and fluorochrome-labelled secondary antibody. Cells were washed in PBS, and the mean intensity of lipofuscin-positive cells in at least 100 cells were counted in random fields in each of the 6 individual repeats using Fiji software.

Oxygen consumption rate measurement. For the oxygen consumption rate (OCR) analysis, PPCs ($\sim 1.5 \times 10^6$) were treated with EVs for 24 h as described above and were harvested with trypsin/EDTA, rinsed with phosphate-buffered saline (PBS) (with 10 % FBS), and centrifuged at $1200 \times g$ for 3 min. Cells were re-suspended in 1.5 ml of PBS. The OCR (nmoles $O_2$/min) was measured for 20 min at 37 °C using a Clark-type electrode (Hansatech). The average OCR was determined from 1 min measurements deriving from four different time points (3, 6, 9, 12 min)[72]. OCR data were normalized to total protein content using the Bradford method.

Immunoblot analysis and antibodies. For western blot analysis, cells were pelleted and tissues from *Er1*<sup>F/−</sup> and *Er1*<sup>F/+</sup> animals were homogenized in Sucrose buffer (0.32 M Sucrose, 15 mM HEPES-KOH, 60 mM KCl, 2 mM EDTA, 0.5 mM EGTA, 0.5%BSA, 0.1%NP-40, pH-7.9 and protease inhibitors). Cell pellets were washed three times with 1x PBS. Cell pellets were then re-suspended in NP-40 lysis buffer (10 mM Tris-HCl pH 7.9, 10 mM NaCl, 3 mM MgCl2, 0.5% NP-40 and protease inhibitors) and incubated for 10 min at 4 °C. The supernatant after centrifugation was kept as the cytoplasmic fraction and the pellets was re-suspended in high-salt extraction buffer (10 mM HEPES–KOH pH 7.9, 380 mM KCl, 3 mM MgCl2, 0.2 mM EDTA, 20% glycerol and protease inhibitors) and incubated for 60 min, 4 °C. The supernatant after centrifugation was kept as the nuclear fraction[95]. For whole-cell extract preparations, cells pellets were resuspended in 150 mM NaCl, 50 mM Tris pH = 7.5, 5% Glycerol, 1% NP-40, 1 mM MgCl) and incubated on ice for 30 min. For western blot analysis of EVs, EV pellets were resuspended in 2x Laemli buffer. Antibodies against Rac2 (C-11, WB: 1:500, IF: 1:100), RhoA (26C4, WB: 1:500, IF: 1/100), CD9 (C-4, WB: 1:500, IF: 1/100), Glut2 (H-67, WB: 1:500, IF: 1:100), Glut3 (B-6, WB: 1:500, IF: 1:100), LC3 (C-9, WB: 1:500, IF: 1:500), FancI (H102, IF: 1:50), Ercc1 (D-10, WB:1:500, IF: 1:50), Albumin (P-20, WB: 1:500, IF: 1/200), Amylase (G-10, WB: 1:500, IF: 1:100), biotin (Rock-land, 600-401-098, IF:1:500), LaminB1 (ab16048, WB:1:1000), p62 (SQSTM1, MBL PM045, WB:1:5000, IF:1:1000), goat anti-rat IgG-CFL 647 (sc-362293, IF: 1:1000) and donkey anti-goat IgG-HRP (sc-2020, WB: 1:5000) were from SantaCruz Biotechnology. γ-H2A.X (05-636, IF: 1:12000), pATM (05–740, IF: 1:100), Rad51 (ABE257, IF: 1:100), Caspase3 (AB3623, IF: 1:200), p-Glut (Ser226) (ABN991), Goat anti-Rabbit IgG Antibody, Peroxidase Conjugated (AP132P, WB: 1:10000) and Goat Anti-Mouse IgG Antibody, Peroxidase Conjugated, H+L (AP124P, WB:

(ORF) of *Rab10 Rac1* and *CD9* genes were amplified by PCR using appropriately designed primers and were incorporated in pEGFP-N1 plasmid fused at the 3′ end with a sequence encoding EGFP, respectively. The primers used were as follows (restriction sites are underlined): Rac1_pEGFPN1_For: GAT<u>CTCGAG</u>ATG CAGGCCATC, Rac1_pEGFPN1_Rev: AGT<u>GGATCC</u>CCAA<u>CAGCAGG</u>, Rab10_pEGFPN1_For: GAT<u>CTCGAG</u>ATG<u>GCGAAGAAG</u>, Rab10_pEGFPN1_Rev: AGT<u>GGATCC</u>CCGCAG<u>GCACTTG</u>, CD9_pEGFPN1_For: GGC<u>CTCGAG</u>ATG CCGGTC<u>AAAGGA</u> CD9_pEGFPN1_Rev: GGG<u>GATCCC</u>CGACC<u>ATTTCTCG</u>. For glucose uptake assay, primary hepatocytes or PPCs were starved in low glucose medium in the presence of EVs for 24 h prior to incubation with 100 μM 2-NBDG (Cat. No.6065, Tocris) for 15–20 min. Cells were washed three times with PBS, fixed and counterstained with DAPI. For mTOR inhibition, PPCs were incubated with the indicated medium and EVs, together with 2 μM Rapamycin (Cat. No 1292, Tocris Bioscience) for 24 h. For SA-β-gal activity a Beta-galactosidase (β-gal) assay kit was used (Abcam Inc. ab65351) according to manufacturer instructions. Briefly differentiated *Er1*<sup>F/+</sup> and *Er1*<sup>F/−</sup> BMDMs were fixed, washed with PBS and stained in β-galactosidase fixative solution at 37 °C until β-gal staining became visible in

1:10000) were from Millipore. GM130 (clone 35, wb: 1:500, IF: 1:200) was from BD Transduction Laboratories. Glut1 (ab40084, WB:1:300, IF: 1:150), Calreticulin (ab2907, IF:1:500), Grp78/BiP (ab21685, WB: 1:500, IF: 1;200), F4/80 (ab6640, WB:1:500), b-tubulin (ab6046, WB:1:1000) and iNOS (ab15323, WB:1:500, IF: 1:100) were from Abcam. Alix (#2171, WB: 1:500), Rab10 (#8127, WB: 1:500, IF: 1:100), NF-κB p65 (#8242, IF: 1:100), p4EBP1 (#2855, WB: 1:500) and Phospho-p70 S6 Kinase (Ser371) (#9208, WB: 1:500) were from Cell Signaling Technology. VCAM (P8B1, IF: 1:200), CD45 (H5A5, IF: 1:200), ICAM (P2A4, IF: 1:100), PECAM (2H8, IF: 1:200) and Mac1 (M1/70.15.11.5.2, IF: 1:200) were from Developmental Studies Hybridoma Bank (DSHB). Rac1 (ARC03, WB:1:500, IF: 1:50) was from Cytoskeleton. PKH67 Green Fluorescent Cell Linker Midi Kit (MIDI67) was from Sigma Aldrich. Goat anti-Mouse IgG (H+L) Cross-Adsorbed Secondary Antibody, Alexa Fluor 488 (A-11001, IF: 1:2000), Goat anti-Mouse IgG (H+L) Cross-Adsorbed Secondary Antibody, Alexa Fluor 555 (A-21422, IF:1:2000), Donkey anti-Rabbit IgG (H+L) Highly Cross-Adsorbed Secondary Antibody, Alexa Fluor 488 (A-21206, IF:1:2000), Donkey anti-Rabbit IgG (H+L) Highly Cross-Adsorbed Secondary Antibody, Alexa Fluor 555 (A-31572, IF:1:2000), Goat anti-Rat IgG (H+L) Cross-Adsorbed Secondary Antibody, Alexa Fluor 555 (A-21434, IF:1:2000) and DAPI (62247, IF:1:20000) were from ThermoFisher.

Flow cytometry. Cohorts of 6–8-months-old, male Er1F/+, Er1F/+ fed on a high fat diet and Er1F/− animals (n = 3) were used. Pancreas, spleen and epididymal fat tissues were minced and digested in 1x PBS/1%BSA/0.1%NaN3/collagenase (2.5 mg/ml) and further processed with a Dounce homogenizer. Red blood cells were lysed in ice cold red blood cell lysis buffer (1.5 M NH4Cl, 0.1 M KHCO3, 0.01 M EDTA). Homogenized tissue was further washed in PBS-BSA buffer and passed through a 100 μM wire mesh. Peripheral blood was isolated with heart puncture and bone marrow was isolated from femurs and tibias. Erythrocytes were lysed as previously mentioned. Samples were further washed in PBS-BSA buffer. Cells were stained with fluorochrome conjugated antibodies (CD45, CD11b, F4/80, CD11c, and CD206, Biolegend) for 20 min at 4 C in PBS/5% FBS. Samples were acquired on a FACS Calibur (BD Biosciences) and analyzed using the FlowJo software (Tree Star).

Mass Spectrometry studies. Er1F/+ and Er1F/− BMDMs were cultured for 24 h. in serum-free medium. The medium was then concentrated using Amicon Ultra-15 Centrifugal Filter Units, resolved on 10% SDS-PAGE gel and stained with Colloidal blue silver (ThermoFisher Scientific, USA). The entire lane was cut out and divided into at least 12 gel plugs, which were each further reduced to 1 mm3 gel pieces and placed in low-bind tubes (Eppendorf UK). Proteins were in-gel digested by using modified trypsin (Roche Diagnostics) in 50 mM ammonium bicarbonate. Peptide mixtures were analyzed by nLC-ESI-MS/MS on a LTQ-Orbitrap XL coupled to an Easy nLC (Thermo Scientific). For the sample preparation and the nLC-ESI-MS/MS analysis, the dried peptides were dissolved in 0.5% formic acid aqueous solution, and the tryptic peptide mixtures were separated on a reversed-phase column (Reprosil Pur C18 AQ, Dr. Maisch GmbH), fused silica emitters 100 mm long with a 75μm internal diameter (ThermoFisher Scientific, USA) packed in-house using a packing bomb (Loader kit SP035, Proxeon). Tryptic peptides were separated and eluted in a linear water-acetonitrile gradient and injected into the MS.

RNA-Seq and Quantitative PCR studies. Total RNA was isolated from Er1F/+ and Er1F/− BMDMs using a Total RNA isolation kit (Qiagen) as described by the manufacturer. For RNA-Seq studies, libraries were prepared using the Illumina® TruSeq® mRNA stranded sample preparation Kit. Library preparation started with 1 μg total RNA. After poly-A selection (using poly-T oligo-attached magnetic beads), mRNA was purified and fragmented using divalent cations under elevated temperature. The RNA fragments underwent reverse transcription using random primers. This is followed by second strand cDNA synthesis with DNA polymerase I and RNase H. After end repair and A-tailing, indexing adapters were ligated. The products were then purified and amplified to create the final cDNA libraries. After library validation and quantification (Agilent 2100 Bioanalyzer), equimolar amounts of all 12 libraries were pooled. The pool was quantified by using the Peqlab KAPA Library Quantification Kit and the Applied Biosystems 7900HT Sequence Detection System. The pool was sequenced by using an Illumina HiSeq 4000 sequencer with a paired-end (2 × 75 cycles) protocol. Quantitative PCR (Q-PCR) was performed with a DNA Engine Opticon device according to the instructions of the manufacturer (MJ Research). The generation of specific PCR products was confirmed by melting curve analysis and gel electrophoresis. Each primer pair was tested with a logarithmic dilution of a cDNA mix to generate a linear standard curve (crossing point (CP) plotted versus log of template concentration), which was used to calculate the primer pair efficiency ($E = 10^{(-1/slope)}$). Hypoxanthine guanine phosphoribosyltransferase1 (Hprt-1) mRNA was used as an external standard. For data analysis, the second derivative maximum method was applied: ($E_{1gene\ of\ interest}$ ΔCP (cDNA of wt. mice - cDNA of Ercc1F/-) gene of interest)/($E_{hprt-1}$ ΔCP (cDNA wt. mice- cDNA) hprt-1). Hprt F: CCCAACATCAACAG GACTCC, Hprt R: CGAAGTGTTGGATACAGGCC, Cxcl5 F:TGCCCCTTCCT CAGTCATAG, Cxcl5 R:GGATCCAGACAGACCTCCTTC, Cxcl24 F:AATTC CAGAAAACCGAGTGG, Cxcl24 R:TGGGCCCCTTTAGAAGGCTGG, Cxcl1 F: CCACACTCAAGAATGGTCGC, Cxcl1 R: GTTGTCAGAAGCCAGCGTTC, Ccl9 F:CCGGGCATCATCTTTATCAG, Ccl9 R:GTCCGTGGTTGTGAGTTTTCC, Cxcl12 F:ACGTCAAGCATCTGAAAATCC, Cxcl12 R:AATTTCGGGCTCAATG CACAC, Csf1 F: CCTCATGAGCAGGAGTATTGC, Csf1 R:AAAGGCAAT CTGGCATGAAG, Ccl7 F:TCCCTGGGAAGCTGTTATCTTC, Ccl7 R: TGGAGTTGGGGTTTTCATGTC, Cxcl13 F:TAGATCGGATTCAAGTTACGC,

Cxcl13 R:GTAACCATTTGCCCACGAGG, Ccl3 F:AGATTCCACGCCAATTCA TC, Ccl3 R:TCAAGCCCCTGCTCTACAC, Ccl5 F:CTCGTGCCCACGTCAAG GAG, Ccl5 R:CCCACTTCTTCTCTGGGTTG, Cxcl9 F:CGGAGATCAAACCT GCCTAG, Cxcl9 R:CTTGAACGACGACGACTTTG.

Data analysis. For Q-PCR data, a two-way t-test was used to extract the statistically significant gene expression data (unless noted otherwise) by means of the IBM SPSS Statistics 19 (IBM, NY, USA), Spotfire (Tibco, CA, USA), Partek (Partek INCoR1porated, MO, USA) and R-statistical package (www.r-project.org/). For mass spectrometry (MS), the MS/MS raw data were loaded in Proteome Discoverer 1.3.0.339 (Thermo Scientific) and run using Mascot 2.3.02 (Matrix Science, London, UK) search algorithm against the Mus musculus theoretical proteome (Last modified 6 July 2015) containing 46,470 entries[96]. A list of common contaminants was included in the database[97]. For protein identification, the following search parameters were used: precursor error tolerance 10 ppm, fragment ion tolerance 0.8 Da, trypsin full specificity, maximum number of missed cleavages 3 and cysteine alkylation as a fixed modification. The resulting.dat and.msf files were subsequently loaded and merged in Scaffold (version 3.04.05, Proteome Software) for further processing and validation of the assigned MS/MS spectra employing PeptideProphet and ProteinProphet algorithms for the identification of proteins[98–100]. Thresholds for protein and peptide identification were set to 99 and 95% accordingly, for proteins with minimum 1 different peptides identified, resulting in a protein false discovery rate (FDR) of <0.1%. For single peptide identifications, we applied the same criteria in addition to manual validation. Protein lists were constructed from the respective peptide lists. For label-free relative quantitation of proteins, we applied a label-free relative quantitation method between the different samples (control versus bait) in order to determine unspecific binders during the affinity purification. All.dat and.msf files created by Proteome Discoverer were merged in Scaffold where label-free relative quantification was performed using the total ion current (TIC) from each identified MS/MS spectra. The TIC is the sum of the areas under all the peaks contained in a MS/MS spectrum and total TIC value results by summing the intensity of the peaks contained in the peak list associated to a MS/MS sample. This approach has advantages in comparison to other label-free methods, such as increased dynamic range and quantification for low spectral counts[101]. Protein lists containing the calculated by Scaffold total TIC quantitative value for each protein ware exported from to Microsoft Excel for further processing. The fold change of protein levels was calculated by dividing the mean total TIC quantitative value in bait samples with the mean value of the control samples for each of the proteins. Proteins having ≥80% protein coverage, ≥1 peptide in each sample (Er1F/− BMDMs) and a fold change ≥1.55 were selected as being significantly enriched in Er1F/− BMDMs compared to Er1F/+ BMDM controls. Significant over-representation of pathways, protein-protein interactions and protein complexes were derived by STRING[102] (http://string-db.org/). For RNA-Seq data analysis, the data were downloaded as FASTQ files and their quality was checked with FASTQC, a quality control tool for high throughput sequence data: http://www.bioinformatics. babraham.ac.uk/projects/fastqc). The data were aligned to mm10 genome assembly available from UCSC via Bowtie2. The differentially expressed genes were identified with metaseqR. Count normalization was performed based on edaseq algorithm. In parallel, cufflinks pipeline (version 2). Significant overrepresentation of pathways and gene networks was determined by Gene Ontology and DAVID (http://david. abcc.ncifcrf.gov/summary.jsp; through BBID, BIOCARTA and KEGG annotations). All experiments were repeated ≥3 times. The data exhibited normal distribution (where applicable). For animal studies, each biological replicate consists of 3–5 mouse tissues or cell cultures per genotype per time point or treatment (unless stated otherwise). None of the samples or animals was excluded from the experiment. The animals or the experiments were not randomized. The investigators were not blinded to allocation during animal experiments and outcome assessment.

**Reporting summary**. Further information on research design is available in the Nature Research Reporting Summary linked to this article.

## Data availability

The mass spectrometry proteomics data have been deposited to the ProteomeXchange Consortium (http://proteomecentral.proteomexchange.org) via the PRIDE partner repository with the dataset identifier PXD015727 and 10.6019/PXD015727. The RNA-Seq. data are deposited in ArrayExpress (https://www.ebi.ac.uk/arrayexpress/), (E-MTAB-8439). All other data and reagents are available from the authors upon reasonable request. The source data underlying Figs. 1–8 and Supplementary Figs. 1–10 are provided as a source data file.

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

## Acknowledgements

The Horizon 2020 ERC Consolidator grant "DeFiNER" (GA64663), the FP7 Marie Curie ITN "aDDRess" (GA316390), "CodeAge" (GA316354), "Marriage" (GA316964), the Horizon 2020 Marie Curie ITN "Chromatin3D (GA GA622934), the Santé Foundation and the ELIDEK grant 1059 supported this work. G.C. is supported by the IKY post-doctoral research fellowship program (MIS: 5001552), co-financed by the European Social Fund- ESF and the Greek government.

## Author contributions

E.G., A.I., G.C., I.P., K.G., M.T., K.S., K.E., P.T., J.D., J.A. performed the experiments and/or analyzed data. V.G.G. generated new reagents. G.A.G. interpreted data and wrote the paper. All relevant data are available from the authors.

## Competing interests

The authors declare no competing interests.
