## [Peer Review File · Nature Communications]

Reviewers' comments:

Reviewer #1 (Remarks to the Author):

The manuscript by Goulielmaki et al examines the effect of persistent DNA damage in tissue-infiltrating macrophages. They demonstrate that DNA damage in Lys2 expressing monocytes, granulocytes and activated macrophages leads to Golgi dispersal, dilation of endoplasmic reticulum, autophagy and increased EV biogenesis. The authors further demonstrate that macrophage-derived EVs accumulate in sera of Ercc1F/- mice. The authors suggest that the EVs confer an increase in the insulin-independent glucose transporter levels, resulting in enhanced cellular glucose uptake and higher cellular oxygen consumption rate. The authors also suggest that the high glucose in EV-targeted cells triggers pro-inflammatory stimuli via mTOR activation. The authors suggest that this pathway results in chronic inflammation and tissue pathology in mice, similar to that observed with aging and progerias. Overall, the results are of interest and are relevant to many aspects of pathologies observed with age and in human progeroid syndromes. However, there are a large number of results in the manuscript which are not fully supported, based more on correlations, derived from analysis of a small number of cells at a given time point. The authors also make sweeping conclusions that are not fully substantiated, especially in vivo, such as that mTOR activation by increased glucose uptake leads to iNOS accumulation and the NF- κ B nuclear translocation with no evidence of mechanism. Determining what effects of the EVs are direct versus indirect requires time course experiments and further use of genetic and pharmacologic approaches. Also, the authors provide no evidence that this effect is specific to macrophages with chronic DNA damage since it is possible that other types of immune or non-immune cells with DNA damage, elevated markers of senescence and increased EV secretion would confer similar effects. Overall, this is an interesting and relevant manuscript that could be improved by additional controls and scaling back of the some of the conclusions.

Specific Comments:

1. Ercc1 is deleted in monocytes, granulocytes and activated macrophages, not just in tissue infiltrating macrophages.
2. Although there are few apoptotic macrophages, it is important to determine if the overall number of macrophages is reduced in the Ercc1F/- mice.
3. Lipofuscin is not an accepted marker of senescence. SA- β gal staining, loss of LaminB, increased p16INK4a and p21Cip1 expression as well as certain SASP markers in macrophages from the Lys2-Cre;Ercc1F/+ and Lys2-Cre;Ercc1F/- mice need to be examined.
4. Senescent cells in general have been shown to drive certain age related pathologies including altering metabolism. Thus the authors need to examine at least one other cell type specific deletion of Ercc1 to demonstrate that the effect is specific to Lys2 positive monocytes/macrophages.
5. The figures and methods provided little detail regarding ages of mice.
6. There are no details regarding how the EVs lead to elevated levels of glucose transporters. Are miRNAs, proteins in addition to Rabs, lipids or metabolites involved?

Reviewer #2 (Remarks to the Author):

In this manuscript, the authors characterize mice in which *Ercc1*, encoding one subunit of an essential DNA repair endonuclease, is deleted “only in tissue infiltrating macrophages” using the lysosomal 2 promoter to drive Cre expression. They claim that this leads to enhanced biogenesis and secretion of vesicles from immune cells. The vesicle content is taken up by other cells leading to altered metabolism, inflammation and mTOR activation.

The work is timely, innovative and of keen interest to a broad audience as a potential mechanism of driving or spreading senescence and aging. DNA damage in immune cells, or potentially any cell, could lead to increased secretory phenomenon, leading to spread of damage and stress signaling and coordinated decline or aging. Thus, the work is important and impactful.

This is tempered with concerns, mostly perhaps because the manuscript was written in a very concise many due possibly to space limits.

The goal or hypothesis of the study was not clearly articulated.

The manuscript is riddled with overinterpretation as well as oversimplification. For example, the statement that *Lys2-Cre* deletes only in tissue-infiltrating macrophages is over simplified. The JAX website states: the line deletes in “myeloid cell lineage, including monocytes, mature macrophages, and granulocytes” The experiments done to bolster the authors statement include measuring *Ercc1* expression in activated macrophages, hepatocytes, primary pancreatic cells and white adipose tissue. This is inadequate to support the authors claim about deletion in one subset of immune cells. This comment is not meant to discredit the quality of the work. Instead it is meant to point out that declarative statements interpreting the data should be 100% accurate rather than narrowly interpreted.

Other examples include using static confocal pictures to support “reversed Golgi dispersal” or “gradual accumulation of intracellular vesicles”. Or an EM image of cytoplasmic filled projections and pseudopodia-like structures containing vesicles supporting claims of enhanced contractile activity of activated macrophages and enhanced formation and secretion of extracellular vesicles.

Virtually every conclusion drawn in this manuscript references a figure panel that illustrates immunofluorescence detection of multiple proteins in 1-2 cells +/- a bar graph quantitating the finding. There is no indication of # mice, # cells, # fields, to know how robust the results and conclusions are. Occasionally, this is accompanied by an immunoblot. But still there is no indication if this is a representative result reflecting multiple mice, tissues, or replicas. Please indicate n values for every experiment to allow a reader to gauge the strength of the data.

Finally, for many conclusions, a single or possibly two markers were measured. For example, lipofuscin was the only measure of senescence. Or confocal detection of LC3b, Gm130 and GRP78 was measured to draw the conclusions that the ER is dilated, Golgi are dispersed, and autophagy is affected. This should be bolstered by measuring other endpoints, or alternative interpretations of the data discussed.

Reviewer #3 (Remarks to the Author):

The authors present work indicating that tissue resident macrophages and those infiltrating tissues from the circulation are vulnerable to DNA damage-associated activation of a senescence phenotype that can be induced by genetic manipulation (Er1 deletion). In this context, the authors show an impact on systemic energy and glucose metabolism, and link this to the production of extracellular vesicles by macrophages, and their metabolic impact on the neighboring cells that may uptake them. The work on extracellular vesicles is interesting, as is the impact demonstrated in controlled systems on the role that these EVs have in regulating glucose uptake. However, there are components that the authors attempt to connect between these in vitro phenotypes and the in vivo phenotype of Er1F/- mice that are still preliminary and contain key mechanistic caveats. These are outlined below.

Major Concerns:

1. Lyz2-Cre is not a highly specific model. It is known that LysM expresses in neurons as well, and this promiscuous expression has been documented in the literature. So, the authors should demonstrate that neuronal populations are not deleted for the gene of interest, especially since they demonstrate a weight phenotype right off the bat.
2. The mice have reduced body weight. They also have improved glucose tolerance when measured at 4 months of diet. Most lean models are glucose tolerant as a result of leanness. Is the glucose tolerance of these mice simply a manifestation of leanness, or can it be dissociated from leanness? This is important, because the authors make a very specific claim based on their in vitro data.
3. Figure 1F is the main figure that the authors use to state that the Er1F/- mice do not have altered insulin sensitivity. Insulin tolerance testing is not sufficient to make this claim. The authors should, at best, clamp these mice, and at the very least measure their tissue insulin-dependent signaling responses and their ability to take up glucose in response to in vivo insulin in tissues, using 2-DOG or similar approach. For example, ITT can be altered because of an alteration in counterregulatory hormone levels and responsiveness as much as it can be altered by insulin sensitivity per se.
4. There is a very important conceptual concern regarding the enhanced glucose tolerance shown in Figure 1, and the claim of a “pro-inflammatory” phenotype shown and discussed later in the paper. Aging is a key driver of senescence and the associated secretory phenotype at the core of this paper. Advanced aging is also associated with increased adiposity and reduced glucose tolerance in mice and people. HFD feeding exacerbates this effect. With age and diet-induced obesity, increasing adiposity and worsening glucose intolerance are associated with inflammation in metabolic tissues. Here, however, senescence induced in myeloid cells genetically is associated with improved glucose tolerance yet increased inflammation. Can the authors reconcile the clear differences between this genetic model (Figure 6 I, for example) and the very different phenotype of myeloid senescence that is seen in the context of advancing age or diet-induced obesity. The inflammation seems to be consistent, but the association with glucose metabolism is opposite to what is seen in wt mice and people.
5. Mechanistically, the paper has two thematically linked but only correlative stories. On one hand, the in vitro work suggests that Er1F/- macrophages make EVs that promote glucose uptake

but induce inflammatory responses in parenchymal cell types that take up the EVs. On the other hand, they have an in vivo story that indicates that Er1F^{-/-} mice have improved glucose tolerance and senescent macrophages in widespread tissues. The authors need to do something more to specifically link the EV story with the in vivo phenotype. Can they block it somehow? Can they target EV uptake in vivo in some way, either pharmacologically or otherwise? Can they block EV secretion?

6. Does aging and myeloid senescence in vivo in wt mice fed a HFD or exposed to any other DNA damage inducing (non-genetic) stimulus also produce EV secretion that leads to a similar functional impact on parenchymal cells, even if tested in cultured cells?

7. Why do the authors rely solely on PPCs, when it is unclear what impact pancreatic cells would have to overall glucose tolerance. By 6 mo of diet, the in vivo glucose tolerance improvement over control (Figure 1 is quite striking). With this, one would think that muscle cells, which are known to contribute heavily to overall glucose clearance, would be a better cell type to study than PPCs. The authors should really study the phenomenon in multiple cell types beyond PPCs, including primary hepatocytes, myotubes and muscle preps, etc.

Minor Concerns:

1. Is the reduced body weight in Er1F^{-/-} mice associated with reduced fat mass, lean mass, or both? Is the length of these mice reduced? It is important to know if this is an anti-obesity phenotype or some other alteration in body mass that could be developmental in nature.

2. The authors on page 5 (initial results section) need to define at their first mention the following acronyms: Ku-55933 (they just call it an inhibitor), ATR/CDK, DNA-PK, and ATM. ATM is very important to define because it commonly stands for adipose tissue macrophages too.

Nature Communications NCOMMS-18-27513-T: "Tissue-infiltrating macrophages mediate an exosome-based metabolic reprogramming upon DNA damage"

Authors' reply to reviewers' remarks

We would like to thank the reviewers for their valuable comments in our work. We have carefully evaluated all the comments and suggestions and performed several additional experiments, which we believe that further strengthen the data and the conclusions presented in our manuscript. Please find below our point-by-point response to the reviewers' remarks. Text changes in the manuscript are highlighted with light grey color.

Reviewers' Comments:

Reviewer #1 (Remarks to the Author):

The manuscript by Goulielmaki et al examines the effect of persistent DNA damage in tissue-infiltrating macrophages. They demonstrate that DNA damage in Lys2 expressing monocytes, granulocytes and activated macrophages leads to Golgi dispersal, dilation of endoplasmic reticulum, autophagy and increased EV biogenesis. The authors further demonstrate that macrophage-derived EVs accumulate in sera of Ercc1F/- mice. The authors suggest that the EVs confer an increase in the insulin-independent glucose transporter levels, resulting in enhanced cellular glucose uptake and higher cellular oxygen consumption rate. The authors also suggest that the high glucose in EV-targeted cells triggers pro-inflammatory stimuli via mTOR activation. The authors suggest that this pathway results in chronic inflammation and tissue pathology in mice, similar to that observed with aging and progerias. Overall, the results are of interest and are relevant to many aspects of pathologies observed with age and in human progeroid syndromes. However, there are a large number of results in the manuscript which are not fully supported, based more on correlations, derived from analysis of a small number of cells at a given time point. The authors also make sweeping conclusions that are not fully substantiated, especially in vivo, such as that mTOR activation by increased glucose uptake leads to iNOS accumulation and the NF- κ B nuclear translocation with no evidence of mechanism. Determining what effects of the EVs are direct versus indirect requires time course experiments and further use of genetic and pharmacologic approaches. Also, the authors provide no evidence that this effect is specific to macrophages with chronic DNA damage since it is possible that other types of immune or non-immune cells with DNA damage, elevated markers of senescence and increased EV secretion would confer similar effects. Overall, this is an interesting and relevant manuscript that could be improved by additional controls and scaling back of some of the conclusions.

Specific Comments:

1. Ercc1 is deleted in monocytes, granulocytes and activated macrophages, not just in tissue infiltrating macrophages.

Authors reply: It was, indeed, not our intention to convince the readership that the Lys-Cre transgene is only expressed in macrophages but that the observed response to EVs is specific to *Er1^{F/-}* macrophages (see our response to point 4 below). We now provide further evidence that the

Lys2-Cre transgene is expressed in monocytes, macrophages and neutrophils. The Lys2-Cre transgene is not expressed in the primary pancreatic cells (also pancreas), primary hepatocytes (also liver), and the white adipose tissue or in neurons. The new data are available in Figure 1 and Figure S1.

2. *Although there are few apoptotic macrophages, it is important to determine if the overall number of macrophages is reduced in the Ercc1F⁻ mice.*

Authors reply: This is a valid point. We now provide with FACS data showing that there is no decrease in the total number of CD45 (+) CD11b (+) macrophages during hematopoiesis (bone marrow) or at peripheral tissues i.e. blood, spleen and pancreas of *Er1^{F/-}* animals as compared to *Er1^{F/+}* corresponding control animals. The new data are provided in Figure S1E-H.

3. *Lipofuscin is not an accepted marker of senescence. SA-βgal staining, loss of LaminB, increased p16INK4a and p21Cip1 expression as well as certain SASP markers in macrophages from the Lys2-Cre;Ercc1F^{+/+} and Lys2-Cre;Ercc1F^{-/-} mice need to be examined.*

Authors reply: Besides lipofuscin, we now provide data on SA-β-gal staining and western blotting on Lamin B1. Similar to the lipofuscin data, we find an increase in beta-galactosidase expression in *Er1^{F/-}* BMDMs (Figure S1C) and a mild but detectable senescence-associated loss of Lamin B1 expression in *Er1^{F/-}* BMDMs (Figure S1D). We also provide evidence for an increase in the mRNA levels of p16INK4a (Cdkn2a), p21Cip1 (Cdkn1a) (Figure 4J) and SASP markers e.g., several chemokines (Figure 4K) and in the protein levels of IL6 and IL8 secreted in the *Er1^{F/-}* macrophage media (Figure 5J).

4. *Senescent cells in general have been shown to drive certain age related pathologies including altering metabolism. Thus the authors need to examine at least one other cell type specific deletion of Ercc1 to demonstrate that the effect is specific to Lys2 positive monocytes/macrophages.*

Authors reply: This is a valid point. To strengthen the specificity of our findings on *Er1^{F/-}* macrophage-derived EVs, we now provide with a set of *in vitro* and *in vivo* experiments. First, we performed studies on primary pancreatic cells exposed to EVs derived from macrophages, adipocytes, mouse embryonic fibroblasts (MEFs) or monocytes with abrogated ERCC1 function. The data reveal an accumulation of GLUT1 and iNOS as well as an enhanced 2-NBDG uptake only in the recipient cells treated with *Er1^{F/-}* macrophage-derived EVs. The new data are provided in Figure S8.

To test for the *in vivo* relevance of our findings, C57BL/6 animals were injected intravenously every 24 hours for a period of 10 days with EVs derived from either wt. or *Er1^{F/-}* MEFs and *Er1^{F/-}* or *Er1^{F/+}* macrophages. Unlike in animals treated with wt. or *Er1^{F/-}* MEF-derived EVs or with *Er1^{F/+}* macrophage-derived EVs, intravenous injection of C57BL/6 animals with *Er1^{F/-}* macrophage-derived EVs, leads to the accumulation of GLUT1, the recruitment of macrophages in the liver and to an enhanced glucose tolerance in these animals. Western blotting further confirmed the specific accumulation of GLUT1 in the liver and muscle protein extracts of animals treated only with *Er1^{F/-}* macrophage-derived EVs. The new data are provided in Figure 8A-E.

5. *The figures and methods provided little detail regarding ages of mice.*

Authors reply: The age of mice is now reported in the figure legends and, where appropriate, in the relevant text of the manuscript.

6. *There are no details regarding how the EVs lead to elevated levels of glucose transporters. Are miRNAs, proteins in addition to Rabs, lipids or metabolites involved?*

Authors reply: This is a rather large topic of intense research in the cell biology/EV field with an ever growing list of RAB, RAS, and RHO family of small GTPases being identified to regulate glucose uptake and/or the trafficking of glucose transporters to cell membrane (e.g. references: 47-50 51,52 53-55 in the manuscript). In this work, we provide evidence that *ErI^{F/-}* macrophage-derived EVs carry several RAB, RAS, and RHO members. To test for the functional relevance of RAB10 and RAC1 in the biogenesis/secretion of EVs in *ErI^{F/-}* macrophages, we transfected *ErI^{F/+}* macrophages with GFP-tagged RAB10 and RAC1. In line with the pronounced accumulation of CD9 and Alix seen in *ErI^{F/-}* macrophages, we were able to detect the marked accumulation of EV-associated protein markers CD9 and Alix in GFP-tagged RAB10/RAC1-transfected macrophages (Figure S4A). To further confirm that macrophage-derived *ErI^{F/-}* EVs also deliver their cargo to recipient PPCs, we also isolated EVs from previously transfected *ErI^{F/-}* macrophages with GFP-tagged RAB10 or RAC1 (Figure S5A). Following the exposure of PPCs to *ErI^{F/-}* EVs carrying the GFP-tagged RAB10 and RAC1 (derived from the previously transfected *ErI^{F/-}* macrophages), we were able to detect both proteins in the cytoplasm of EV-recipient PPCs (Figure 6B and C).

Reviewer #2 (Remarks to the Author):

*In this manuscript, the authors characterize mice in which *Ercc1*, encoding one subunit of an essential DNA repair endonuclease, is deleted “only in tissue infiltrating macrophages” using the lysosomal 2 promoter to drive Cre expression. They claim that this leads to enhanced biogenesis and secretion of vesicles from immune cells. The vesicle content is taken up by other cells leading to altered metabolism, inflammation and mTOR activation.*

The work is timely, innovative and of keen interest to a broad audience as a potential mechanism of driving or spreading senescence and aging. DNA damage in immune cells, or potentially any cell, could lead to increased secretory phenomenon, leading to spread of damage and stress signaling and coordinated decline or aging. Thus, the work is important and impactful.

This is tempered with concerns, mostly perhaps because the manuscript was written in a very concise many due possibly to space limits.

The goal or hypothesis of the study was not clearly articulated.

Authors reply: The working hypothesis and the overall conclusions supported from our findings are discussed in the first and last paragraphs of the discussion section.

The manuscript is riddled with overinterpretation as well as oversimplification. For example, the

statement that Lys2-Cre deletes only in tissue-infiltrating macrophages is over simplified. The JAX website states: the line deletes in “myeloid cell lineage, including monocytes, mature macrophages, and granulocytes” The experiments done to bolster the authors statement include measuring Ercc1 expression in activated macrophages, hepatocytes, primary pancreatic cells and white adipose tissue. This is inadequate to support the authors claim about deletion in one subset of immune cells. This comment is not meant to discredit the quality of the work. Instead it is meant to point out that declarative statements interpreting the data should be 100% accurate rather than narrowly interpreted.

Authors reply: This is a valid point. It was, indeed, not our intention to convince the readership that the Lys-Cre transgene is only expressed in macrophages but that the observed response to EVs is specific to macrophages. We now provide evidence that the Lys2-Cre transgene is expressed in monocytes, macrophages and neutrophils. The Lys2-Cre transgene is not expressed in the primary pancreatic cells (also pancreas), primary hepatocytes (also liver), and the white adipose tissue or in neurons. The new data are available in Figure 1 and Figure S1.

To strengthen the specificity of our findings on $ErI^{F/-}$ macrophage-derived EVs, we now provide with a set of *in vitro* and *in vivo* experiments on i. primary pancreatic cells exposed to $ErI^{F/+}$ or $ErI^{F/-}$ EVs derived from macrophages or monocytes and to EVs derived from $ErI^{-/-}$ or wt. adipocytes or MEFs as well as ii. C57BL/6 animals that are injected intravenously with EVs derived from either wt. or $ErI^{-/-}$ MEFs and $ErI^{F/+}$ or $ErI^{F/-}$ macrophages. The new data are provided in Figure 8 and S8.

Other examples include using static confocal pictures to support “reversed Golgi dispersal” or “gradual accumulation of intracellular vesicles”. Or an EM image of cytoplasmic filled projections and pseudopodia-like structures containing vesicles supporting claims of enhanced contractile activity of activated macrophages and enhanced formation and secretion of extracellular vesicles.

Authors reply: We now provide with live confocal imaging data in $ErI^{F/-}$ and $ErI^{F/+}$ macrophages transiently expressing CD9-GFP. Unlike the sedentary appearance of $ErI^{F/+}$ macrophages, we evidenced the gradual formation of a new vesicle-like structure in the cytoplasm, a fusion event of a cytoplasmic vesicle-like structure with the plasma membrane and the presence of newly emerging pseudopodia in the membrane of $ErI^{F/-}$ macrophages. The new data are provided in Figure S4B, S4C, S4D and Video files S1-S4.

We further provide evidence for the relevance of RAB10 and RAC1 (besides of other GTPases) in the biogenesis/secretion of EVs and the uptake of $ErI^{F/-}$ EVs carrying such GTPases in recipient cells. First, we transfected $ErI^{F/+}$ macrophages with GFP-tagged RAB10 and RAC1. In line with the pronounced accumulation of CD9 and Alix in $ErI^{F/-}$ macrophages, we detect the marked accumulation of EV-associated protein markers in GFP-tagged RAB10/RAC1-transfected macrophages. The new data are provided in Figure S4A.

Next, we transfect $ErI^{F/-}$ macrophages with GFP-tagged RAB10 or RAC1 (Figure S5A). Following the exposure of PPCs to $ErI^{F/-}$ EVs carrying the GFP-tagged RAB10 and RAC1, we were able to

detect both proteins in the cytoplasm of EV-recipient PPCs (Figure 6B and C). These data confirm that macrophage-derived $Er1^{F/-}$ EVs deliver their cargo to recipient PPCs.

Virtually every conclusion drawn in this manuscript references a figure panel that illustrates immunofluorescence detection of multiple proteins in 1-2 cells +/- a bar graph quantitating the finding. There is no indication of # mice, # cells, # fields, to know how robust the results and conclusions are. Occasionally, this is accompanied by an immunoblot. But still there is no indication if this is a representative result reflecting multiple mice, tissues, or replicas. Please indicate n values for every experiment to allow a reader to gauge the strength of the data.

Authors reply: Figure legends now provide with the n values on the number of mice, tissues, cells or biological replicates tested.

Finally, for many conclusions, a single or possibly two markers were measured. For example, lipofuscin was the only measure of senescence. Or confocal detection of LC3b, Gm130 and GRP78 was measured to draw the conclusions that the ER is dilated, Golgi are dispersed, and autophagy is affected. This should be bolstered by measuring other endpoints, or alternative interpretations of the data discussed.

Authors reply: For senescence, we now provide, in addition to lipofuscin, with data on SA- β -gal staining and western blotting on Lamin B1. The new data show an increase in β -galactosidase expression in $Er1^{F/-}$ BMDMs (Figure S1C) and a mild but detectable senescence-associated loss of Lamin B1 expression in $Er1^{F/-}$ BMDMs (Figure S1D). We also provide evidence for an increase in the mRNA levels of senescence-associated markers p16INK4a (Cdkn2a), p21Cip1 (Cdkn1a) (Figure 4J) and SASP markers e.g. several chemokines (Figure 4K) and in the protein levels of IL6 and IL8 secreted in the $Er1^{F/-}$ macrophage media (Figure 5J).

For autophagy, we now provide, in addition to LC3, with confocal microscopy and western blot data on P62, a reporter for autophagic activity.

To further validate GRP78, P62, LC3 and GM130 as relevant biomarkers for the observed cytoplasmic stress responses in $Er1^{F/-}$ macrophages, we now provide with western blotting and/or confocal microscopy studies in $Er1^{F/+}$ BMDMs treated with i. nocodazole (known to trigger Golgi dispersal), ii. chloroquine (known to inhibit the degradation of autophagosomes in lysosomes) and iii. tunicamycin (known to trigger ER stress). Overall, the results confirm the validity of these markers for the observed ER stress, Golgi dispersal and autophagy seen in $Er1^{F/-}$ macrophages (Figure 2A-B). The new data are provided in Figure S2A and S2B.

Reviewer #3 (Remarks to the Author):

The authors present work indicating that tissue resident macrophages and those infiltrating tissues from the circulation are vulnerable to DNA damage-associated activation of a senescence phenotype that can be induced by genetic manipulation (Er1 deletion). In this context, the authors

show an impact on systemic energy and glucose metabolism, and link this to the production of extracellular vesicles by macrophages, and their metabolic impact on the neighboring cells that may uptake them. The work on extracellular vesicles is interesting, as is the impact demonstrated in controlled systems on the role that these EVs have in regulating glucose uptake. However, there are components that the authors attempt to connect between these in vitro phenotypes and the in vivo phenotype of $Er1F^{-/-}$ mice that are still preliminary and contain key mechanistic caveats. These are outlined below.

Major Concerns:

1. $Lyz2-Cre$ is not a highly specific model. It is known that $LysM$ expresses in neurons as well, and this promiscuous expression has been documented in the literature. So, the authors should demonstrate that neuronal populations are not deleted for the gene of interest, especially since they demonstrate a weight phenotype right off the bat.

Authors reply: It was, indeed, not our intention to convince the readership that the $Lys-Cre$ transgene is only expressed in macrophages but that the observed response to EVs is specific to $Er1^{F/-}$ macrophages (please, see our response below). We now provide further evidence that the $Lys2-Cre$ transgene is expressed in monocytes, macrophages and neutrophils. The $Lys2-Cre$ transgene is not expressed in the primary pancreatic cells (also pancreas), primary hepatocytes (also liver), and the white adipose tissue or in neurons. The new data are available in Figure 1 and Figure S1.

2. The mice have reduced body weight. They also have improved glucose tolerance when measured at 4 months of diet. Most lean models are glucose tolerant as a result of leanness. Is the glucose tolerance of these mice simply a manifestation of leanness, or can it be dissociated from leanness? This is important, because the authors make a very specific claim based on their in vitro data.

Authors reply: This is a valid point. We now provide with further evidence that, in spite of the fact, that the 2-months old $Er1^{F/-}$ and $Er1^{F/+}$ animals fed on a high-fat diet show marked differences in GTT (Figure 3C), these animals have no significant differences in body weight (Figure S2C). The reverse is also true: the 2-months old $Er1^{F/-}$ and $Er1^{F/+}$ animals fed for only 2 months on normal diet show detectable differences in weight (Figure 3A) but no differences in GTT when compared to the age-matched $Er1^{F/+}$ animals (Figure 3B; as shown). These findings uncouple $Er1^{F/-}$ leanness from the observed differences in GTT.

3. Figure 1F is the main figure that the authors use to state that the $Er1F^{-/-}$ mice do not have altered insulin sensitivity. Insulin tolerance testing is not sufficient to make this claim. The authors should, at best, clamp these mice, and at the very least measure their tissue insulin-dependent signaling responses and their ability to take up glucose in response to in vivo insulin in tissues, using 2-DOG or similar approach. For example, ITT can be altered because of an alteration in counterregulatory hormone levels and responsiveness as much as it can be altered by insulin sensitivity per se.

Authors reply: This is a valid point. We now provide further evidence that overnight starved $Er1^{F/-}$ and $Er1^{F/+}$ animals manifest comparable levels of 2-Deoxy-D-glucose (2-DG) uptake in the liver and muscle protein extracts in response to insulin (Figure S2D). In line with the enhanced glucose tolerance previously seen in $Er1^{F/-}$ animals, we also find that 2-DG uptake is significantly higher in the tissues of non-insulin treated $Er1^{F/-}$ mice when compared to $Er1^{F/+}$ corresponding controls (Figure S2D).

4. There is a very important conceptual concern regarding the enhanced glucose tolerance shown in Figure 1, and the claim of a “pro-inflammatory” phenotype shown and discussed later in the paper. Aging is a key driver of senescence and the associated secretory phenotype at the core of this paper. Advanced aging is also associated with increased adiposity and reduced glucose tolerance in mice and people. HFD feeding exacerbates this effect. With age and diet-induced obesity, increasing adiposity and worsening glucose intolerance are associated with inflammation in metabolic tissues. Here, however, senescence induced in myeloid cells genetically is associated with improved glucose tolerance yet increased inflammation. Can the authors reconcile the clear differences between this genetic model (Figure 6 I, for example) and the very different phenotype of myeloid senescence that is seen in the context of advancing age or diet-induced obesity. The inflammation seems to be consistent, but the association with glucose metabolism is opposite to what is seen in wt mice and people.

Authors reply: This is a truly interesting observation. In this work, we provide evidence that DNA damage triggers an exosome-based, metabolic reprogramming that leads to chronic inflammation. As DNA damage accumulates with aging and nearly all progeroid syndromes originate from inborn defects in genome maintenance, we can only speculate that DNA damage accumulation with advancing age plays a role in the response. Importantly, we find that the response is restricted to EVs derived only from $Er1^{F/-}$ macrophages. We have, therefore, no evidence to assume that all senescent cells (or recipient cells) would respond in a similar manner or what the outcome will be when –unlike in $Er1^{F/-}$ animals- all cells are DNA repair-deficient or all cells indiscriminately accumulate DNA damage with age. Besides senescence, DNA damage also triggers cancer whose incidence also increases with aging. Cancer cells rely on increased glucose intake. We are currently embarking on a new set of experiments to test whether the response to macrophage-derived EVs is associated with enhanced cancer incidence in $Er1^{F/-}$ animals over time and/or with aging.

5. Mechanistically, the paper has two thematically linked but only correlative stories. On one hand, the in vitro work suggests that $Er1^{F/-}$ macrophages make EVs that promote glucose uptake but induce inflammatory responses in parenchymal cell types that take up the EVs. On the other hand, they have an in vivo story that indicates that $Er1^{F/-}$ mice have improved glucose tolerance and senescent macrophages in widespread tissues. The authors need to do something more to specifically link the EV story with the in vivo phenotype. Can they block it somehow? Can they target EV uptake in vivo in some way, either pharmacologically or otherwise? Can they block EV secretion?

Authors reply: Indeed, this is an excellent point. To strengthen the specificity of our findings on $Er1^{F/-}$ macrophage-derived EVs, we now provide with a set of *in vitro* and *in vivo* experiments.

First, we performed studies on primary pancreatic cells exposed to EVs derived from macrophages, adipocytes, mouse embryonic fibroblasts (MEFs) or monocytes with abrogated ERCC1 function. The data reveal the accumulation of GLUT1 and iNOS as well as an enhanced 2-NBDG uptake only in the recipient cells treated with *Er1^{F/-}* macrophage-derived EVs. The new data are provided in Figure S8.

To test for the *in vivo* relevance of our findings, C57BL/6 animals were injected intravenously every 24 hours for a period of 10 days with EVs derived from either wt. or *Er1^{F/-}* MEFs and *Er1^{F/+}* or *Er1^{F/-}* macrophages. Unlike in animals treated with wt. or *Er1^{F/-}* MEF-derived EVs or with *Er1^{F/+}* macrophage-derived EVs, intravenous injection of C57BL/6 animals with *Er1^{F/-}* macrophage-derived EVs, leads to the accumulation of GLUT1, the recruitment of macrophages in the liver and to an enhanced glucose tolerance in these animals. Western blotting further confirmed the specific accumulation of GLUT1 in the liver and muscle protein extracts of animals treated only with *Er1^{F/-}* macrophage-derived EVs. The new data are provided in Figure 8A-E.

6. Does aging and myeloid senescence in vivo in wt mice fed a HFD or exposed to any other DNA damage inducing (non-genetic) stimulus also produce EV secretion that leads to a similar functional impact on parenchymal cells, even if tested in cultured cells?

Authors reply: This is a valid point. We provide with new data revealing a similar accumulation of CD9 and Alix in the EV fraction of *Er1^{F/-}* monocytes and neutrophils and a mild or negligible accumulation of CD9 and Alix in *Er1^{F/-}* primary mouse embryonic fibroblasts (MEFs) and the adipocytes, respectively (when each cell type is compared to the corresponding control cells) (Figure S3E). Our data also reveal differences in EV secretion between MEFs, monocytes, neutrophils or the adipocytes as evidenced by the relative CD9 and Alix protein levels in the EV fraction of wt. and *Er1^{F/+}* control cells. Based on the available literature and our findings, we reason, therefore, that several cell types, including those of parenchymal origin, are, in principle, able to secrete EVs upon DNA damage either due to a DNA repair defect or after exposure to genotoxins. However, we provide with *in vitro* and *in vivo* evidence that the impact of *Er1^{F/-}* macrophage-derived EVs in recipient cells is specific (see our previous response on the specificity of *Er1^{F/-}* macrophage-derived EVs in cells or animals).

7. Why do the authors rely solely on PPCs, when it is unclear what impact pancreatic cells would have to overall glucose tolerance. By 6 mo of diet, the in vivo glucose tolerance improvement over control (Figure 1 is quite striking). With this, one would think that muscle cells, which are known to contribute heavily to overall glucose clearance, would be a better cell type to study than PPCs. The authors should really study the phenomenon in multiple cell types beyond PPCs, including primary hepatocytes, myotubes and muscle preps, etc.

Authors reply: Indeed, muscle cells are known to contribute to overall glucose clearance. In this work, we opted for primary pancreatic cells (PPCs) or the hepatocytes as both cell types are relevant for coupling growth stimuli with fine-tuning mechanisms involved in nutrient-sensing and glucose homeostasis. We now provide with further evidence showing that the muscle and liver of

overnight starved $Er1^{F/-}$ animals manifest an increase in 2-Deoxy-D-glucose (2-DG) uptake when compared to the corresponding tissues of $Er1^{F/+}$ animals (Figure S2D). We also provide with western blot data further confirming the accumulation of GLUT1 in muscle and liver protein extracts of C57BL/6 animals intravenously injected only with $Er1^{F/-}$ macrophage-derived EVs (Figure 8C-D).

Minor Concerns:

1. Is the reduced body weight in $Er1^{F/-}$ mice associated with reduced fat mass, lean mass, or both? Is the length of these mice reduced? It is important to know if this is an anti-obesity phenotype or some other alteration in body mass that could be developmental in nature.

Authors reply: We now provide with data showing that the reduced body weight of $Er1^{F/-}$ animals associates with reduced fat mass and that adult $Er1^{F/-}$ animals have comparable body size (nasoanal length) to age-matched $Er1^{F/+}$ animals.

2. The authors on page 5 (initial results section) need to define at their first mention the following acronyms: Ku-55933 (they just call it an inhibitor), ATR/CDK, DNA-PK, and ATM. ATM is very important to define because it commonly stands for adipose tissue macrophages too.

Authors reply: We now define ATM as “Ataxia-telangiectasia mutated (Atm)”, ATR as “Ataxia telangiectasia and Rad3 related”, DNA-PK as “DNA-dependent protein kinase” and Ku-55933 as a competitive ATM inhibitor.

We wish to really thank all three reviewers for their insightful remarks which we believe have significantly improved the impact, overall scope and significance of our work.

Reviewers' comments:

Reviewer #2 (Remarks to the Author):

The authors have done an excellent job of responding to the critiques. In particular, the inclusion of further analysis of senescence markers and injection of WT mice with EVs from Ercc1-deficient macrophages have significantly strengthened the manuscript. The authors also have tried to tone down their conclusions. Thus my only remaining concern is in regard to the specificity of macrophage EVs in conferring the observed effects. Although the authors have shown that EVs from Ercc1-deficient MEFs are not able to confer the same effects as macrophage EVs in culture and in vivo, the authors have not adequately addressed whether the observed effects are indeed due to macrophage EVs specifically or to EVs from other types of senescent or damaged immune/non immune cells. This could be examined quickly in cell culture experiments.

Reviewer #3 (Remarks to the Author):

The authors have done a lot to improve their manuscript, and the new studies add to the depth and completeness of the paper. However, a few key issues remain unresolved, and these detract from the impact and scope of the study. These are straightforward and addressable.

1. Despite assessing gene expression in primary neurons (they do not provide details on the area of the brain or age of the mice), the authors still do not know if Cre-Lox recombination has taken place among neurons in the hypothalamus, nor do they have any idea as to the mechanism of the body weight phenotype. It is known that LyzM-Cre is expressed in neurons (e.g. Orthgeiss, et al. *Eu. J. Immunol.* 2016; 46(6): 1529-32). So, it would be ideal to show no recombination in the arcuate nucleus or whole hypothalamus, for example by crossing Lyz2-Cre mice with a fluorescent reporter and assessing histochemistry. Do Er1 F/- mice eat more than controls? If so, then there is a need to at least rule out a direct effect in the hypothalamus.
2. Alternatively, it is possible (especially if the KO mice do NOT eat more) that the exosome pathway to increase glucose uptake shunts glucose away from adipose tissue (where it might be stored as triglyceride). It may instead be consumed in other tissues, leading to increased energy consumption and negative energy balance. To probe this, the authors could look at primary cells by Seahorse XF for example, to show that Er1 KO macrophage-produced exosomes increase oxygen consumption in these recipient cells (PPCs, hepatocytes, myocytes, etc). This would at least support the contention that the glucose consumed is being constitutively burned. This could be supportive of the weight loss phenotype.

3. The authors did not really respond to initial comment #4. The point is that chronic inflammation in the white adipose tissue is consistently associated with insulin resistance and elevated blood sugar. Moreover, advancing age, which also triggers senescence, does the same. Here, however, the authors claim they can “force” senescence via DNA damage through Er1 deletion. In this context, they see adipose tissue inflammation (expected), but IMPROVED glucose homeostasis (not expected). So, two questions need to be answered:

a. What is the actual polarization state of the adipose macrophages in these mice? Are they different from WT mice fed a HFD, in which the polarization state has been well characterized as more pro-inflammatory? It is possible that Er1 knockout induces an accumulation of macrophages that are polarized in a direction different from those models associated with insulin resistance.

b. We are assuming that the genetic knockout induces DNA damage accumulation. Although senescence markers are shown, is there any direct demonstration of accumulated DNA damage in a gene dose-associated manner?

These two pieces of data would really shore up the paper.

4. The authors show that the Er1 KO macrophages produce exosomes that have an effect to increase glucose uptake, in vitro and in vivo. This is a powerful observation. But it is not clear that this production of exosomes is a function of DNA damage, only the genetic deletion. So, the authors should show that macrophages in which DNA damage is induced some other way also make exosomes that improve glucose uptake in other cells. If so, then DNA damage is supported as a core mechanism. If not, then the exosome phenotype is specific to Er1 KO, but not necessarily DNA damage.

5. Do the EV-treated mice have any body weight phenotype? If so, then it really indicates that the macrophage-specific EVs produce all of the phenotype (reducing the chance that any neuronal gene deletion plays a role in the original mouse model). If not, then it helps separate the EV effect from the mechanism leading to weight loss.

Reviewer #4 (Remarks to the Author):

Although the authors have made an effort to answer certain queries, there is still a number of major points that remain unresolved.

Major Concerns;

“However, there are a large number of results in the manuscript which are not fully supported, based more on correlations, derived from analysis of a small number of cells at a given time point. The authors also make sweeping conclusions that are not fully substantiated (...)”

We believe that this problem remains in the current version of the manuscript. Importantly, many figures are showing only one cell or a very small number of cells (e.g.: Fig. 1 A, B, C, F, J; 2A, B, C, E, F, G, H, I; Fig. 3G, I; Fig. 4 A, B, D, E, F); Fig. 5 F, H, I, K, L, M; Fig. 6 A, B, C, G, I, K, L; Fig. 7 F, G); Fig. 8 A, B, C, D) and there is no data quantification that supports the conclusions made by the authors.

1. *Ercc1* is deleted in monocytes, granulocytes and activated macrophages, not just in tissue infiltrating macrophages.

The authors have clearly shown that there is expression of the *Lys2-Cre* transgene in other cell types apart from macrophages. In addition, macrophage-derived EVs have been successfully shown to have an effect in vitro and in BL6 injected mice. However, we want to stress that there can still exist a contribution to the phenotype observed in the mice deriving from myeloid cells different from macrophages. Thus, as long as it is confirmed that the results shown by the authors are robust enough to support the conclusions, we believe that this point has been reasonably addressed.

2. Although there are few apoptotic macrophages, it is important to determine if the overall number of macrophages is reduced in the *Ercc1F/-* mice.

This point has been addressed.

3. Lipofuscin is not an accepted marker of senescence. SA- β gal staining, loss of LaminB, increased p16INK4a and p21Cip1 expression as well as certain SASP markers in macrophages from the *Lys2-Cre;Ercc1F/+* and *Lys2-Cre;Ercc1F/-* mice need to be examined.

There is a lack of quantification and statistical significance in the results presented to support this key point. Beta-gal staining (which is the most accepted senescence marker) quantification is not statistically significant and the one-lane image of lamin B1 WB is neither convincing nor quantified.

4. Senescent cells in general have been shown to drive certain age related pathologies including altering metabolism. Thus the authors need to examine at least one other cell type specific deletion of *Ercc1* to demonstrate that the effect is specific to *Lys2* positive monocytes/macrophages.

We agree that this point made by the Reviewer is key. The in vitro experiments are convincing as long as the data provided are proven to be robust (i.e. showing proper quantifications). We consider that the in vivo experiment consisting on the injection of EVs obtained either from macrophages or MEFs into mice and testing GLUT1 expression, numbers of infiltrating macrophages and glucose tolerance could have served to elegantly address this point. However, the experiments shown are lacking any quantification so that it is not possible for us to evaluate whether the images shown are representative. In addition, we feel that n=3 mice is quite a low number of animals to support any conclusion.

5. The figures and methods provided little detail regarding ages of mice.

This point has been improved.

6. There are no details regarding how the EVs lead to elevated levels of glucose transporters. Are miRNAs, proteins in addition to Rabs, lipids or metabolites involved?

The transfection of Rab10-GFP and Rac1-GFP constructs does not allow to draw any conclusions about their role in biogenesis/secretions of EVs, as it serves only as a reporter of the presence of EVs. Experiments inhibiting those pathways would have been more informative.

Minor points:

Language editing would be desirable. Additionally, there are some typos and mistakes throughout the manuscript:

- “possitve”
- Fig. 1.E: The meaning of “W”, “N”, and “C” (presumably Whole, Nuclei and Cytoplasm) is not referred in the figure legend.
- Fig. 1.E: Loading control is missing.
- Scale bars are missing in most of the figures.
- Fig. 2.I: The vertical line on the left side delimiting the genotypes is not correct.
- Fig. S.8: Adipocytes and MEFs are supposed to be obtained from Er1^{-/-} instead of Er1F^{-/-} mice.
- Legend to Fig. S8.A: iNOS expression is not shown by immunofluorescence as it stated in the legend, but by WB.

Nature Communications NCOMMS-18-27513-A: "Tissue-infiltrating macrophages mediate an exosome-based metabolic reprogramming upon DNA damage"

Authors' reply to reviewers' remarks

We would like to thank the reviewers for their remaining comments on our work. Please find below our point-by point response to the remaining reviewers remarks. Text changes in the manuscript are highlighted with light grey color.

Reviewers' Comments:

Reviewers' comments

Reviewer #2: *The authors have done an excellent job of responding to the critiques. In particular, the inclusion of further analysis of senescence markers and injection of WT mice with EVs from Ercc1-deficient macrophages have significantly strengthened the manuscript. The authors also have tried to tone down their conclusions. Thus my only remaining concern is in regard to the specificity of macrophage EVs in conferring the observed effects. Although the authors have shown that EVs from Ercc1-deficient MEFs are not able to confer the same effects as macrophage EVs in culture and in vivo, the authors have not adequately addressed whether the observed effects are indeed due to macrophage EVs specifically or to EVs from other types of senescent or damaged immune/non immune cells. This could be examined quickly in cell culture experiments.*

Authors' Reply: We thank the reviewer for her/his kind comment on the improvement of our manuscript. In addition to the *in vivo* experiments shown in **Figure 8**, we show that the observed effects are specifically triggered by $Er1^{F/-}$ macrophage-derived EVs. Indeed, the treatment of primary pancreatic cells with EVs derived from either $Er1^{F/+}$ or $Er1^{F/-}$ adipocytes, MEFs, or monocytes had no significant effect on GLUT1 or iNOS accumulation (**Figure S10A**) or on the glucose uptake in PPCs (**Figure S10B**).

Reviewer #3: *The authors have done a lot to improve their manuscript, and the new studies add to the depth and completeness of the paper. However, a few key issues remain unresolved, and these detract from the impact and scope of the study. These are straightforward and addressable.*

1. Despite assessing gene expression in primary neurons (they do not provide details on the area of the brain or age of the mice), the authors still do not know if Cre-Lox recombination has taken place among neurons in the hypothalamus, nor do they have any idea as to the mechanism of the body weight phenotype. It is known that LyzM-Cre is expressed in neurons (e.g. Orthgeiss, et al. Eu. J. Immunol. 2016; 46(6): 1529-32). So, it would be ideal to show no recombination in the arcuate nucleus or whole hypothalamus, for example by crossing Lyz2-Cre mice with a fluorescent reporter and assessing histochemistry.

Authors' reply: We thank their reviewer for her/his kind comment on the improvement of our manuscript. With respect to the remaining concerns:

-In **Figure S1**, we provide data on whole brain tissues derived from 4-months old animals. In **Figure S3**, we now provide with data on the hypothalamus of 7-months old animals.

-To test whether Cre-Lox recombination has taken place among neurons in the hypothalamus, we first performed western blot and immunofluorescence studies for ERCC1 protein in the hypothalamus of $Er1^{F/+}$ or $Er1^{F/-}$ mice and find no detectable differences (**Figure S3C** and **Figure S3D; left panel**, as indicated). Next, we intercrossed *Lyz2-Cre* mice with Rosa26-YFP mice and find only the minimal detection of YFP signal colocalized with the neuronal marker NeuN in Rosa26-YFP^{*Lyz2-Cre*} hypothalamus (**Figure S3D; right panel**). Indeed, in Rosa26-YFP^{*Lyz2-Cre*} animals, the great majority of YFP colocalized with MAC1. To further strengthen our findings, we also examined the accumulation of γ -H2AX foci in the hypothalamus of $Er1^{F/+}$ or $Er1^{F/-}$ mice. The γ -H2AX histone variant is a well-established DNA damage marker that routinely accumulates in ERCC1-defective cells (due to the accumulation of irreparable DNA lesions). In line with our previous findings, we find no accumulation of γ H2AX foci in $Er1^{F/-}$ hypothalamic nuclei indicating that this tissue is proficient in DNA repair (**Figure S3D; left panel**, as indicated).

Do Er1 F/- mice eat more than controls? If so, then there is a need to at least rule out a direct effect in the hypothalamus.

Authors' reply: A daily measurement of food intake (over a period of two weeks) in $Er1^{F/-}$ and $Er1^{F/+}$ animals did not reveal any significant differences (**Figure S3B**).

2. Alternatively, it is possible (especially if the KO mice do NOT eat more) that the exosome pathway to increase glucose uptake shunts glucose away from adipose tissue (where it might be stored as triglyceride). It may instead be consumed in other tissues, leading to increased energy consumption and negative energy balance. To probe this, the authors could look at primary cells by Seahorse XF for example, to show that Er1 KO macrophage-produced exosomes increase oxygen consumption in these recipient cells (PPCs, hepatocytes, myocytes, etc). This would at least support the contention that the glucose consumed is being constitutively burned. This could be supportive of the weight loss phenotype.

Authors' Reply: Indeed, we find that exposure of primary cells to EVs derived from $Er1^{F/-}$ cells or $Er1^{F/-}$ sera or wt. cells exposed to the genotoxin mitomycin trigger an increase in oxygen consumption rate. The data are shown in **Figure 6J**. Based on these and our previous findings, we, therefore, agree with the Reviewer that the weight reduction in $Er1^{F/-}$ animals likely originates from the substantial increase in glucose uptake, increased oxygen consumption rate and the negative energy balance in these animals.

3. The authors did not really respond to initial comment #4. The point is that chronic inflammation in the white adipose tissue is consistently associated with insulin resistance and elevated blood sugar. Moreover, advancing age, which also triggers senescence, does the same. Here, however, the authors claim they can "force" senescence via DNA damage through Er1 deletion. In this context, they see adipose tissue inflammation (expected), but IMPROVED glucose homeostasis (not expected). So, two questions need to be answered:

a. What is the actual polarization state of the adipose macrophages in these mice? Are they different from WT mice fed a HFD, in which the polarization state has been well characterized as more pro-inflammatory? It is possible that *Er1* knockout induces an accumulation of macrophages that are polarized in a direction different from those models associated with insulin resistance.

Authors' Reply: To address the reviewer's remark, we conducted a series of flow cytometry studies on the epididymal white adipose tissue of *Er1*^{F/-} mice and *Er1*^{F/+} mice fed on a high fat diet (*Er1*^{F/+} HFD). Our FACS analysis revealed an increase in the infiltration of F4/80 (+) and M1 (+) macrophage infiltrates in the epididymal fat of *Er1*^{F/-} and *Er1*^{F/+} fed on a high fat diet (*Er1*^{F/+} HFD) animals (**Figure S4A-B and S4D**). Unlike with *Er1*^{F/+} HFD mice, however, we also find the relative increase of the M2 (+) population in the white adipose macrophages in *Er1*^{F/-} mice (**Figure S4D**, as indicated). This finding indicates that the ERCC1 defect triggers the accumulation of macrophages that are polarized in a direction distinct to that seen in animal models associated with insulin resistance.

b. We are assuming that the genetic knockout induces DNA damage accumulation. Although senescence markers are shown, is there any direct demonstration of accumulated DNA damage in a gene dose-associated manner? These two pieces of data would really shore up the paper.

Authors' Reply: We have tested several DNA damage markers i.e. γ -H2AX, FANCI, pATM, RAD51 to show that DNA repair-deficient *Ercc1*^{-/-} cells accumulate DNA damage. Our analysis revealed that the number of γ -H2AX (+) nuclei is significantly higher in the DNA repair-defective *Er1*^{F/-} BMDMs (**Figure 1G**) and TEMs (**Figure 1H**) compared to wt. cells. We also find marked differences in the number of positively stained nuclei for FANCI involved in the repair of DNA interstrand crosslinks, RAD51 involved in the repair of DNA double strand breaks by homologous recombination and phosphorylated ATM, a central mediator of the DNA damage response (**Figure 1I** for BMDMs and **Figure 1J** for TEMs). Finally, we also find that the dose-dependent accumulation of DNA damage marker γ -H2AX coincides with the EV secretion in *Er1*^{+/+}, *Er1*^{+/-} and *Er1*^{-/-} macrophages (**Figure S5D**).

4. The authors show that the *Er1* KO macrophages produce exosomes that have an effect to increase glucose uptake, in vitro and in vivo. This is a powerful observation. But it is not clear that this production of exosomes is a function of DNA damage, only the genetic deletion. So, the authors should show that macrophages in which DNA damage is induced some other way also make exosomes that improve glucose uptake in other cells. If so, then DNA damage is supported as a core mechanism. If not, then the exosome phenotype is specific to *Er1* KO, but not necessarily DNA damage.

Authors' Reply: We provide several experiments that show that exposure of wt. cells to a genotoxic agent (i.e. mitomycin which is denoted as "MMC") triggers the secretion of EVs and that the "MMC-EVs" trigger similar responses to those observed when recipient cells are treated with EVs derived from *Er1*^{F/-} cells. The data are shown in **Figures 5I, 6F, 6J, 7A, 7C, 7E, 7G, 7F, 7G, 7H, 8D, 8E and 9A**.

5. Do the EV-treated mice have any body weight phenotype? If so, then it really indicates that the macrophage-specific EVs produce all of the phenotype (reducing the chance that any neuronal gene deletion plays a role in the original mouse model). If not, then it helps separate the EV effect from the mechanism leading to weight loss.

Authors' Reply: In this work, mice have been treated with EVs for only a short time period i.e. 10 days, which makes it unlikely to result in any meaningful weight differences. In the revised manuscript, we provide with data that show no detectable differences in (i). the hypothalamic expression of ERCC1, (ii). the food intake and (iii). the accumulation of DNA damage-associated γ H2AX foci in the hypothalamus, thereby, minimizing the chance that a putative neuronal gene deletion plays a role in the $Er1^{F/-}$ mouse model.

Reviewer #4: *Although the authors have made an effort to answer certain queries, there is still a number of major points that remain unresolved.*

Major Concerns;

“However, there are a large number of results in the manuscript which are not fully supported, based more on correlations, derived from analysis of a small number of cells at a given time point. The authors also make sweeping conclusions that are not fully substantiated (...)”

We believe that this problem remains in the current version of the manuscript. Importantly, many figures are showing only one cell or a very small number of cells (e.g.: Fig. 1 A, B, C, F, J; 2A, B, C, E, F, G, H, I; Fig. 3G, I; Fig. 4 A, B, D, E, F); Fig. 5 F, H, I, K, L, M; Fig. 6 A, B, C, G, I, K, L; Fig. 7 F, G); Fig. 8 A, B, C, D) and there is no data quantification that supports the conclusions made by the authors.

Authors' Reply: In certain figures, the larger magnification of images helps the readership to visualize the immunofluorescence signal that would have been impossible to appreciate on photos with a smaller magnification and especially in Figures where a great number of images (e.g. ~30 images) are shown (e.g. Figure 1 and elsewhere).

We now provide with data quantification for all figures either as an inlay representation of the average % of (+) stained cells +/- standard error of the mean (typically for confocal microscopy or immunohistochemistry images) or as graphs (typically for western blots or immunohistochemistry images). Due to space limitations, the data quantification graphs for certain images are provided in supplemental figures (in that case, this is mentioned in the relevant figure legend). Specifically, data quantification are now provided as inlay numbers for **Figures 1A, 1B, 1F, 2A, 2B, 2D, 2E, 2F, 2G, 4C, 4D, 4E, 4F, 5K, 5L, 5M, 6A, 6B, 6C, 6I, 6L, 8A, 8B** and as graphs for **Figures 1E, 1J, 2C, 2H, 3G, 3I, 4A, 4B, 5F (shown in S5A), 5H (shown in S5A), 5I (shown in S5C), 6G (shown in S7E), 6K (shown in S8A), 7F, 7G, 8C and 8D**. Data quantification are not provided for Figure 1C and 1D where the signal is completely absent in the control or the tested sample, as well as for Figure 2I, where the electron microscopy images are descriptive single macrophage cell views from $Er1^{F/+}$ or $Er1^{F/-}$ animals (n=3) and data cannot be quantified in a meaningful way. However, the derived conclusions have been independently confirmed on subsequent experiments supporting the EV biogenesis and secretion upon DNA damage or in $Er1^{F/-}$ macrophages (Figure 5F-K, Figure S2A-B, Figure S5, Figure S6, Figure S7 and Figure S10).

1. *Ercc1* is deleted in monocytes, granulocytes and activated macrophages, not just in tissue infiltrating macrophages. The authors have clearly shown that there is expression of the *Lys2-Cre* transgene in other cell types apart from macrophages. In addition, macrophage-derived EVs have been successfully shown to have an effect *in vitro* and in BL6 injected mice. However, we want to stress that there can still exist a contribution to the phenotype observed in the mice deriving from myeloid cells different from macrophages. Thus, as long as it is confirmed that the results shown by the authors are robust enough to support the conclusions, we believe that this point has been reasonably addressed.

Authors' Reply: We thank the reviewer for her/his comment. Indeed, we provide *in vivo* and *in vitro* evidence that the observed effects are specific to *Er1^{F/-}* macrophage-derived EVs. The *in vivo* data are shown in **Figure 8A-B**. The *in vitro* data are shown in **Figure S8**.

2. Although there are few apoptotic macrophages, it is important to determine if the overall number of macrophages is reduced in the *Ercc1F/-* mice. This point has been addressed.

Authors' Reply: We thank the reviewer for this comment.

3. Lipofuscin is not an accepted marker of senescence. SA- β gal staining, loss of LaminB, increased *p16INK4a* and *p21Cip1* expression as well as certain SASP markers in macrophages from the *Lys2-Cre;Ercc1F/+* and *Lys2-Cre;Ercc1F/-* mice need to be examined. There is a lack of quantification and statistical significance in the results presented to support this key point. Beta-gal staining (which is the most accepted senescence marker) quantification is not statistically significant and the one-lane image of lamin B1 WB is neither convincing nor quantified.

Authors' Reply: The statistically significant accumulation of lipofuscin is shown in **Figure 1L**. The statistically significant upregulation for *p16INK4a* and *p21Cip1* mRNA levels (RNA-Seq studies) is shown in **Figure 4J**. The statistically significant increase of SA- β -gal (+) cells in *Er1^{F/-}* samples is now provided as an inlay percentage +/- standard error of the mean in **Figure S1C**. The statistically significant decrease in Lamin B1 protein levels is provided as a graph in **Figure S1D**.

4. Senescent cells in general have been shown to drive certain age related pathologies including altering metabolism. Thus the authors need to examine at least one other cell type specific deletion of *Ercc1* to demonstrate that the effect is specific to *Lys2* positive monocytes/macrophages. We agree that this point made by the Reviewer is key. The *in vitro* experiments are convincing as long as the data provided are proven to be robust (i.e. showing proper quantifications). We consider that the *in vivo* experiment consisting on the injection of EVs obtained either from macrophages or MEFs into mice and testing GLUT1 expression, numbers of infiltrating macrophages and glucose tolerance could have served to elegantly address this point. However, the experiments shown are lacking any quantification so that it is not possible for us to evaluate whether the images shown are representative. In addition, we feel that $n=3$ mice is quite a low number of animals to support any conclusion.

Authors' Reply: We provide *in vivo* and *in vitro* evidence that the observed effects are specific to *Er1^{F/-}* macrophage-derived EVs. The *in vivo* data are shown in **Figure 8A-B**. The *in vitro* data are shown in **Figure S8**. We now also provide with quantification data as an inlay percentage +/- standard error of the mean in **Figure 8A** and **8B** or as a graph for the western blot images shown in

Figure 8C and 8D. For the GTT measurements, we have now increased the original number of EV-treated animals (n=3) to 8 animals per group (for those animals treated with macrophage-derived EVs) and to 6 animals per group (for those animals treated with MEFs-derived EVs).

5. *The figures and methods provided little detail regarding ages of mice. This point has been improved.*

Authors' Reply: We thank the reviewer for this comment.

6. *There are no details regarding how the EVs lead to elevated levels of glucose transporters. Are miRNAs, proteins in addition to Rabs, lipids or metabolites involved? The transfection of Rab10-GFP and Rac1-GFP constructs does not allow to draw any conclusions about their role in biogenesis/secretions of EVs, as it serves only as a reporter of the presence of EVs. Experiments inhibiting those pathways would have been more informative.*

Authors' Reply: Indeed, transfection of macrophages with GFP-tagged RAB10 and RAC1 has been used as a reporter for the presence of EVs in **Figure 6B** and **Figure 6C**. However, we also show that the overexpression of RAB10 and RAC1 in macrophages leads to increased biogenesis/secretion of EVs as evidenced by the marked accumulation of EV-associated protein markers CD9 and Alix (**Figure S6A**). To further meet the Reviewer's remark, we have now treated *Er1^{F/-}* macrophages with the RAC1 inhibitor NSC 23766 for 16 hrs. We find that inhibition of RAC1 activity in *Er1^{F/-}* macrophages leads to a decrease in EV secretion, as evidenced by the reduction in CD9 protein levels (**Figure S6B**).

Minor points:

Language editing would be desirable. Additionally, there are some typos and mistakes throughout the manuscript:

- "positve"

Authors' Reply: This has now been corrected.

- *Fig. 1.E: The meaning of "W", "N", and "C" (presumably Whole, Nuclei and Cytoplasm) is not referred in the figure legend.*

Authors' Reply: This has now been corrected.

- *Fig. 1.E: Loading control is missing.*

Authors' Reply: The Tubulin and Fibrillarin protein levels now serve as loading controls.

- *Scale bars are missing in most of the figures.*

Authors' Reply: This has now been corrected.

- *Fig. 2.I: The vertical line on the left side delimiting the genotypes is not correct.*

Authors' Reply: This has now been corrected.

- *Fig. S.8: Adipocytes and MEFs are supposed to be obtained from *Er1^{-/-}* instead of *Er1^{F/-}* mice.*

Authors' Reply: This has now been corrected.

- *Legend to Fig. S8.A: iNOS expression is not shown by immunofluorescence as it stated in the legend, but by WB.*

Authors' Reply: This has now been corrected.